# Bio-xLSTM: Generative modeling, representation and in-context learning of biological and chemical sequences

**Niklas Schmidinger**[1]    **Lisa Schneckenreiter**[1]    **Philipp Seidl**[1]    **Johannes Schimunek**[1]
**Pieter-Jan Hoedt**[1]    **Johannes Brandstetter**[1,2]    **Andreas Mayr**[1]
**Sohvi Luukkonen**[1]    **Sepp Hochreiter**[1,2]    **Günter Klambauer**[1,2]

[1] ELLIS Unit Linz and LIT AI Lab, Institute for Machine Learning,
Johannes Kepler University, Linz, Austria
[2] NXAI GmbH, Linz, Austria

## Abstract

Language models for biological and chemical sequences enable crucial applications such as drug discovery, protein engineering, and precision medicine. Currently, these language models are predominantly based on Transformer architectures. While Transformers have yielded impressive results, their quadratic runtime dependency on the sequence length complicates their use for long genomic sequences and in-context learning on proteins and chemical sequences. Recently, the recurrent xLSTM architecture has been shown to perform favorably compared to Transformers and modern state-space model (SSM) architectures in the natural language domain. Similar to SSMs, xLSTMs have a linear runtime dependency on the sequence length and allow for constant-memory decoding at inference time, which makes them prime candidates for modeling long-range dependencies in biological and chemical sequences. In this work, we tailor xLSTM towards these domains and propose a suite of architectural variants called Bio-xLSTM. Extensive experiments in three large domains, genomics, proteins, and chemistry, were performed to assess xLSTM's ability to model biological and chemical sequences. The results show that models based on Bio-xLSTM a) can serve as proficient generative models for DNA, protein, and chemical sequences, b) learn rich representations for those modalities, and c) can perform in-context learning for proteins and small molecules.

## 1 Introduction

**Accurate computational models for biological sequences are essential for translating data into actionable insights in modern biology.** Biological sequences like DNA, RNA, and proteins are central to molecular biology, genomics, and drug discovery. Major projects like the Human Genome Project (Lander et al., 2001) and the 1000 Genomes Project (1000 Genomes Project Consortium, 2010) have driven large-scale data collection efforts. Modeling these sequences is key to advancing life sciences (Benegas et al., 2023; Karollus et al., 2024), interacting with biological systems (Hopf et al., 2017; Riesselman et al., 2018; Yang et al., 2019) or predicting phenotypes from genetic variants (Ashley, 2016; Brandes et al., 2023; Acosta et al., 2022). Similar efforts exist for protein sequences (The UniProt Consortium, 2023) and small molecules (Kim et al., 2023; Zdrazil et al., 2023), used for tasks like protein engineering (Arnold, 2018; Yang et al., 2019), predicting 3D structures (Jumper et al., 2021), and drug discovery (Zhavoronkov et al., 2019). Large language models (LLMs) (Brown et al., 2020; Bubeck et al., 2023) have emerged as prime candidates for modeling biological sequences and serving as foundation models for molecular biology and chemistry (Ji et al., 2021; Schiff et al., 2024; Nguyen et al., 2023; Rives et al., 2021; Lin et al., 2023).

**Large language models for biological sequences must handle long sequences and incorporate context.** The rise of LLMs (Radford et al., 2018; Brown et al., 2020; Bubeck et al., 2023) has

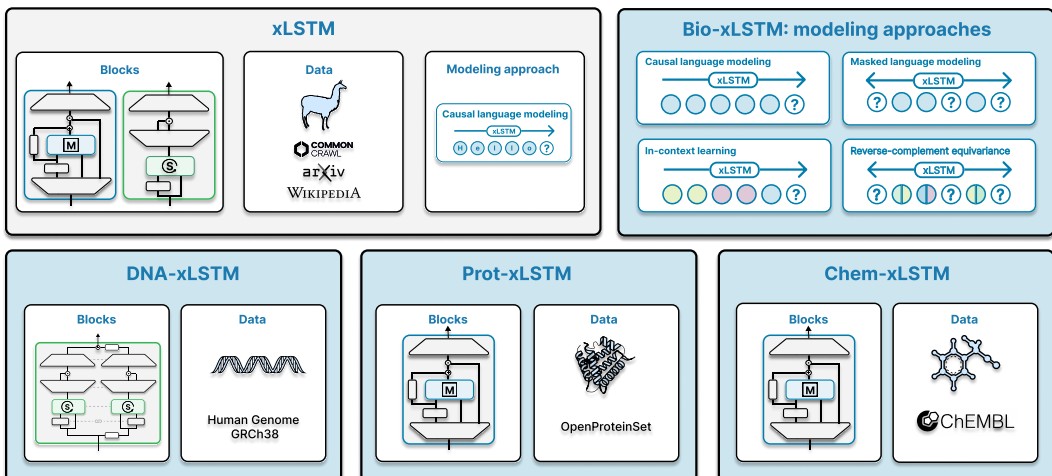

Figure 1: Overview of Bio-xLSTM. **Top left:** xLSTM for natural language processing tasks. **Top right:** Considered modeling approaches for biological sequences: masked language modeling, equivariance to reverse complementary sequence, and in-context learning. **Bottom left:** DNA-xLSTM models are trained on genomic DNA sequences and then fine-tuned on downstream tasks. **Bottom center:** Prot-xLSTM models are trained in a causal modeling setting with a fill-in-the-middle objective and use homologous proteins for in-context learning. **Bottom right:** Chem-xLSTM models are trained to generate small molecules. For an in-context learning setting, Chem-xLSTM models use molecules with known properties as the context.

revolutionized numerous fields, including life sciences. Most LLMs are based on the Transformer architecture (Vaswani et al., 2017), which excels at predicting the next or missing token using self-attention. However, the self-attention mechanism scales quadratically with sequence length, making long-sequence modeling computationally expensive. As a result, most biological sequence models use short contexts (Rives et al., 2021; Ji et al., 2021; Dalla-Torre et al., 2024). However, biological sequences require long context windows for accurate modeling because of their important long-range interactions due to 3D folding (Anfinsen, 1973), or gene regulation in DNA (Bouwman and de Laat, 2015). The human genome spans around three billion base-pairs (bps), far exceeding the context limits of Transformer-based models. Furthermore, long contexts also benefit models that exploit homologous proteins (Truong Jr and Bepler, 2023; Sgarbossa et al., 2024) and molecular context for small molecules (Papadatos et al., 2010; Schimunek et al., 2023). The emergence of state-space models (SSMs), like S4 (Gu et al., 2022), Hyena (Poli et al., 2023), and Mamba (Gu and Dao, 2024), enables handling longer contexts in biological domains (Nguyen et al., 2023; Schiff et al., 2024; Sgarbossa et al., 2024). However, the recently proposed xLSTM architecture (Beck et al., 2024), a recurrent neural network, has outperformed SSM architectures in natural language processing. For further related work, see Appendix Section A.

**The recently proposed xLSTM is a powerful architecture for sequence modeling and a promising candidate for biological and chemical sequences.** The xLSTM architecture (Beck et al., 2024) introduces enhanced memory structures and exponential gates that boost its performance, particularly in natural language modeling. Despite these enhancements over traditional LSTM, xLSTM retains the efficiency of a recurrent neural network and can handle varying sequence lengths effectively (Beck et al., 2024), while maintaining expressivity (Merrill et al., 2024) and scalability (Katharopoulos et al., 2020; Choromanski et al., 2021). These features make xLSTM ideal for modeling: i) DNA sequences, which are inherently long and for which long-range interactions between distant parts of the sequence have been observed, ii) protein sequences, where modeling strongly benefits from contextual information of evolutionary-related proteins (Rives et al., 2021), and iii) small molecules represented as chemical sequences, such as Simplified Molecular Input Line Entry System (SMILES) (Weininger, 1988), for which in-context learning (ICL) abilities are an option to generate new molecules with desired properties or from a particular molecular domain (Segler et al., 2018; Schimunek et al., 2023). However, it remains unclear how to best tailor xLSTM for biological and chemical sequences and how xLSTM compares to other domain-specific LLMs architectures.

We introduce: a) DNA-xLSTM, an architectural variant tailored for DNA sequences with reverse-complement equivariant blocks, and evaluate its performance on long-context generative modeling, representation learning, and downstream tasks. b) Prot-xLSTM, a variant for homology-aware protein language modeling with ICL for both generative modeling and inpainting, which we benchmark on generative modeling, conditioned protein design and protein variant fitness prediction tasks. c) Chem-xLSTM, an architectural variant for SMILES representations of small molecules for which we demonstrate ICL capabilities. An overview of Bio-xLSTM is shown in Figure 1.

## 2 BACKGROUND AND NOTATION

xLSTM (Beck et al., 2024) consists of two distinct layer types: sLSTM (Section 2.1) and mLSTM (Section 2.2). These layers are the components within the block structures (Section 2.3) of its multi-layer architectures. We consider a series of input vectors $\boldsymbol{x}_t \in \mathbb{R}^D$ given at a certain time step $t \in \{1, \ldots, T\}$. $\boldsymbol{X} = \boldsymbol{X}_{1:T} = (\boldsymbol{x}_1, \boldsymbol{x}_2, \ldots, \boldsymbol{x}_T) \in \mathbb{R}^{D \times T}$ denotes the matrix of stacked input vectors from all time steps. Both sLSTM and mLSTM are recurrent neural networks, which either map a state $(\boldsymbol{h}_{t-1}, \boldsymbol{c}_{t-1}, \boldsymbol{n}_{t-1})$ to a successor state $(\boldsymbol{h}_t, \boldsymbol{c}_t, \boldsymbol{n}_t)$ given an input $\boldsymbol{x}_{t-1}$ (sLSTM) or a state $(\boldsymbol{h}_{t-1}, \boldsymbol{C}_{t-1}, \boldsymbol{n}_{t-1})$ to a successor state $(\boldsymbol{h}_t, \boldsymbol{C}_t, \boldsymbol{n}_t)$ given an input $\boldsymbol{x}_{t-1}$ (mLSTM). Here, $\boldsymbol{h}_t \in \mathbb{R}^d$ denotes a hidden state, $\boldsymbol{c}_t \in \mathbb{R}^d$ and $\boldsymbol{C}_t \in \mathbb{R}^{d \times d}$ denote cell states responsible for long-term memory and, $\boldsymbol{n}_t \in \mathbb{R}^d$ denotes a normalizer state. sLSTM and mLSTM utilize several adjustable weight matrices and bias vectors (detailed equations below) and employ input-, output-, and forget-gates, activated by exponential (exp) or the sigmoid functions ($\sigma$). For cell inputs in sLSTM, the hyperbolic tangent function (tanh, abbreviated as $\varphi$) is used as an activation function.

### 2.1 SLSTM

The forward pass of sLSTM in the vectorized version is defined as follows:

$$\boldsymbol{c}_t = \mathbf{f}_t \odot \boldsymbol{c}_{t-1} + \mathbf{i}_t \odot \boldsymbol{z}_t \qquad \text{cell state} \qquad (1)$$

$$\boldsymbol{n}_t = \mathbf{f}_t \odot \boldsymbol{n}_{t-1} + \mathbf{i}_t \qquad \text{normalizer state} \qquad (2)$$

$$\boldsymbol{h}_t = \mathbf{o}_t \odot \tilde{\boldsymbol{h}}_t \,, \qquad \tilde{\boldsymbol{h}}_t = \boldsymbol{c}_t \odot \boldsymbol{n}_t^{-1} \qquad \text{hidden state} \qquad (3)$$

$$\boldsymbol{z}_t = \varphi\left(\tilde{\boldsymbol{z}}_t\right) \,, \qquad \tilde{\boldsymbol{z}}_t = \boldsymbol{W_z}\, \boldsymbol{x}_t + \boldsymbol{R_z}\, \boldsymbol{h}_{t-1} + \boldsymbol{b_z} \qquad \text{cell input} \qquad (4)$$

$$\mathbf{i}_t = \exp\left(\tilde{\mathbf{i}}_t\right) \,, \qquad \tilde{\mathbf{i}}_t = \boldsymbol{W_i}\, \boldsymbol{x}_t + \boldsymbol{R_i}\, \boldsymbol{h}_{t-1} + \boldsymbol{b_i} \qquad \text{input gate} \qquad (5)$$

$$\mathbf{f}_t = \exp\left(\tilde{\mathbf{f}}_t\right) \text{ OR } \sigma\left(\tilde{\mathbf{f}}_t\right) \,, \qquad \tilde{\mathbf{f}}_t = \boldsymbol{W_f}\, \boldsymbol{x}_t + \boldsymbol{R_f}\, \boldsymbol{h}_{t-1} + \boldsymbol{b_f} \qquad \text{forget gate} \qquad (6)$$

$$\mathbf{o}_t = \sigma\left(\tilde{\mathbf{o}}_t\right) \,, \qquad \tilde{\mathbf{o}}_t = \boldsymbol{W_o}\, \boldsymbol{x}_t + \boldsymbol{R_o}\, \boldsymbol{h}_{t-1} + \boldsymbol{b_o} \qquad \text{output gate,} \qquad (7)$$

where $\mathbf{i}_t, \mathbf{o}_t, \mathbf{f}_t \in \mathbb{R}^d$ are the input, output and forget gates, respectively, $\boldsymbol{W_z}, \boldsymbol{W_i}, \boldsymbol{W_f}, \boldsymbol{W_o} \in \mathbb{R}^{d \times D}$, $\boldsymbol{R_z}, \boldsymbol{R_i}, \boldsymbol{R_f}, \boldsymbol{R_o} \in \mathbb{R}^{d \times d}$, and $\boldsymbol{b_z}, \boldsymbol{b_i}, \boldsymbol{b_f}, \boldsymbol{b_o} \in \mathbb{R}^d$ are trainable weight matrices and biases, respectively.

### 2.2 MLSTM

The forward pass of the mLSTM is defined as follows:

$$\boldsymbol{C}_t = \mathrm{f}_t\, \boldsymbol{C}_{t-1} + \mathrm{i}_t\, \boldsymbol{v}_t\, \boldsymbol{k}_t^\top \qquad \text{cell state} \qquad (8)$$

$$\boldsymbol{n}_t = \mathrm{f}_t\, \boldsymbol{n}_{t-1} + \mathrm{i}_t\, \boldsymbol{k}_t \qquad \text{normalizer state} \qquad (9)$$

$$\boldsymbol{h}_t = \mathbf{o}_t \odot \tilde{\boldsymbol{h}}_t \,, \qquad \tilde{\boldsymbol{h}}_t = \boldsymbol{C}_t \boldsymbol{q}_t \,/\, \max\left\{\left|\boldsymbol{n}_t^\top \boldsymbol{q}_t\right|, 1\right\} \qquad \text{hidden state} \qquad (10)$$

$$\boldsymbol{q}_t = \boldsymbol{W_q}\, \boldsymbol{x}_t + \boldsymbol{b_q} \qquad \text{query input} \qquad (11)$$

$$\boldsymbol{k}_t = \frac{1}{\sqrt{d}} \boldsymbol{W_k}\, \boldsymbol{x}_t + \boldsymbol{b_k} \qquad \text{key input} \qquad (12)$$

$$\boldsymbol{v}_t = \boldsymbol{W_v}\, \boldsymbol{x}_t + \boldsymbol{b_v} \qquad \text{value input} \qquad (13)$$

$$\mathrm{i}_t = \exp\left(\tilde{\mathrm{i}}_t\right) \,, \qquad \tilde{\mathrm{i}}_t = \boldsymbol{w}_\mathrm{i}^\top\, \boldsymbol{x}_t + b_\mathrm{i} \qquad \text{input gate} \qquad (14)$$

$$\mathrm{f}_t = \sigma\left(\tilde{\mathrm{f}}_t\right) \text{ OR } \exp\left(\tilde{\mathrm{f}}_t\right), \qquad \tilde{\mathrm{f}}_t = \boldsymbol{w}_\mathrm{f}^\top\, \boldsymbol{x}_t + b_\mathrm{f} \qquad \text{forget gate} \qquad (15)$$

$$\mathbf{o}_t = \sigma\left(\tilde{\mathbf{o}}_t\right) \,, \qquad \tilde{\mathbf{o}}_t = \boldsymbol{W_o}\, \boldsymbol{x}_t + \boldsymbol{b_o} \qquad \text{output gate} \qquad (16)$$

where $i_t \in \mathbb{R}$ and $f_t \in \mathbb{R}$ are now scalar input and forget gates, $\boldsymbol{q}_t, \boldsymbol{k}_t, \boldsymbol{v}_t \in \mathbb{R}^d$ are query, key and value inputs with trainable weight matrices $\boldsymbol{W}_q, \boldsymbol{W}_k, \boldsymbol{W}_v \in \mathbb{R}^{d \times D}$, $\boldsymbol{w}_i, \boldsymbol{w}_f \in \mathbb{R}^D$ are input and forget gate weights and the respective $b_i, b_f \in \mathbb{R}$ biases. All other quantities are identical to sLSTM.

## 2.3 BLOCK STRUCTURES

The sLSTM and mLSTM layers are integrated into larger residual backbones (Srivastava et al., 2015; He et al., 2016), which incorporate layer normalization (Ba et al., 2016), pre- or post-up projection layers (Vaswani et al., 2017; Dao, 2024), with short causal convolutions and group normalization (Wu and He, 2020). Figure A1 depicts sLSTM and mLSTM blocks, as well as, a bidirectional mLSTM configuration with weight-tied blocks. For more details, refer to (Beck et al., 2024, Sec.2.4). For Bio-xLSTM we retain these basic building blocks but adjust them to the respective domains. The entire architecture, including all layers, normalization, blocks, and other components, defines a mapping from an input sequence of length $t$ to an output sequence. This mapping is denoted as $\mathrm{xLSTM} : \mathbb{R}^{D \times t} \mapsto \mathbb{R}^{D \times t}$, where xLSTM transforms the stacked inputs up to time step $t$, i.e., $\boldsymbol{X}_{1:t} := (\boldsymbol{x}_1, \boldsymbol{x}_2, \ldots, \boldsymbol{x}_t) \in \mathbb{R}^{D \times t}$, to the corresponding stacked outputs of sequence length $t$, i.e., $\boldsymbol{Y}_{1:t} := (\boldsymbol{y}_1, \boldsymbol{y}_2, \ldots, \boldsymbol{y}_t) \in \mathbb{R}^{D \times t}$ [1]. The $i$-th sequence element is denoted with the subscript $i$, e.g., the $i$-th element from $\boldsymbol{X}_{1:t}$ would be $(\boldsymbol{X}_{1:t})_i$. Similarly to the mapping xLSTM, we also define mappings for the sequence-wise input-/output behavior of layers themselves for an sLSTM layer ($\mathrm{sLSTM} : \mathbb{R}^{D \times t} \mapsto \mathbb{R}^{D \times t}$) or an mLSTM layer ($\mathrm{mLSTM} : \mathbb{R}^{D \times t} \mapsto \mathbb{R}^{D \times t}$). If the specific parameters used for the mapping are unclear, we will denote this by including a second argument in the function, separated by a semicolon. See Appendix Section B.1 for details on the block structures.

## 2.4 MODES OF OPERATION: PARALLEL, CHUNKWISE, AND RECURRENT

The recurrent forms of sLSTM and mLSTM, introduced in Sections 2.1 and 2.2, provide efficient, constant-memory decoding during inference. This eliminates the need for expensive key-value caching, which represents a major challenge for Transformer models in long-range settings. Like Transformers, mLSTM allows for parallelization across the sequence length, significantly speeding up training. Additionally, similar to linear attention variants (Katharopoulos et al., 2020; Yang et al., 2024), mLSTM supports chunkwise parallel processing, blending recurrent and parallel modes (Beck et al., 2025). This approach is especially advantageous for long-sequence training and prompt encoding. For further details, refer to Appendix Section B.2.

## 3 BIO-XLSTM: LONG-RANGE MODELING OF BIOLOGICAL AND CHEMICAL SEQUENCES

Bio-xLSTM introduces three xLSTM-based architectural variants tailored specifically to DNA (Section 3.4), proteins (Section 3.5) and small molecules (Section 3.6). For these application domains, we extend xLSTM from causal language modeling (CLM) (Section 3.1) to new modeling approaches such as fill-in the middle (FIM), in-context learning (ICL) and masked language modeling (MLM) (Section 3.2).

## 3.1 CAUSAL LANGUAGE MODELING AND NEXT-TOKEN PREDICTION

Causal language modeling (CLM), which is the xLSTM's default paradigm, uses the

$$\text{CLM loss: } \mathcal{L}^{\mathrm{CLM}} = \mathbb{E}_{\boldsymbol{X} \sim p_{\boldsymbol{X}}} \, \mathbb{E}_{t \sim [[1, T-1]]} \, \mathrm{CE} \left( \boldsymbol{x}_{t+1}, \mathrm{xLSTM}(\boldsymbol{X}_{1:t})_t \right), \tag{17}$$

where CE is the cross-entropy loss (with logits), $p_{\boldsymbol{X}}$ is the data distribution, and $[[1, T-1]]$ is the discrete uniform distribution from 1 to $T-1$. The objective measures how well a particular sequence token $\boldsymbol{x}_{t+1}$ can be predicted based on the previous tokens $\boldsymbol{X}_{1:t}$ by the model $\mathrm{xLSTM} : \mathbb{R}^{D \times t} \mapsto \mathbb{R}^{D \times t}$. Therefore, this type of modeling is sometimes also called *next token prediction (NTP)*, *uni-directional modeling* or *autoregressive (AR) modeling* and the loss is also called *NTP loss*.

**Fill-in the middle (FIM)** is a modeling paradigm that integrates aspects of both CLM and MLM (Bavarian et al., 2022). In this approach, parts of the sequence are replaced by mask tokens, and for

---

[1]Here $\boldsymbol{x}_i$ and $\boldsymbol{y}_i$, respectively, represent the inputs to and outputs from a particular model from an instance of an xLSTM architecture, rather than the inputs and outputs of a specific sLSTM or mLSTM layer

each replaced segment, both the mask token and the corresponding masked sequence are appended to the end of the sequence. This allows the model to utilize the entire context to predict the masked tokens while maintaining an AR training framework. Consequently, the model can perform both i) generative modeling and ii) inpainting with CLM.

**In-context learning (ICL)** is a capability of language models to learn and perform tasks by leveraging additional information provided as the contextual input without updating their parameters (Brown et al., 2020; Min et al., 2022). This approach allows models to learn from analogy, drawing insights from patterns in the context to adapt their behavior. When the contextual input sequence is denoted as $\boldsymbol{Z} \in \mathbb{R}^{D \times S}$ and the conventional input sequence is denoted as $\boldsymbol{X} \in \mathbb{R}^{D \times T}$, then with an ICL model, we obtain $\boldsymbol{Y}' = \text{xLSTM}([\boldsymbol{Z}, \boldsymbol{X}])_{(S+1):(S+T)}$, where $[\boldsymbol{Z}, \boldsymbol{X}]$ indicates concatenation of contextual and conventional inputs, and, the subscript $(S+1):(S+T)$ indicates that the last output tokens (those corresponding to the $\boldsymbol{X}$ tokens) are selected. For natural language processing tasks, the context $\boldsymbol{Z}$ often contains the solution to a similar problem or exemplary solutions that guide the input and inform the model. For biological and chemical sequences, the context $\boldsymbol{Z}$ could represent similar genetic regions, homologous proteins, or molecules with desired properties, enabling the model to make informed predictions or generate outputs based on these analogies.

## 3.2 MASKED LANGUAGE MODELING (MLM)

Bio-xLSTM extends xLSTM to masked modeling of biological sequences, for which the typical de-masking or de-noising objective (Vincent et al., 2010; Devlin et al., 2019) is used, concretely the

$$\text{MLM loss: } \mathcal{L}^{\text{MLM}} = \mathbb{E}_{\boldsymbol{X} \sim p_{\boldsymbol{X}}} \mathbb{E}_{t \sim [[1,T]]} \mathbb{E}_{\boldsymbol{M} \sim p_{\boldsymbol{M}}} \text{CE}\left(\boldsymbol{x}_t, \text{xLSTM}(\boldsymbol{X} \odot \boldsymbol{M})_t\right), \quad (18)$$

where $\boldsymbol{M} \in \{0,1\}^{D \times T}$ is a random matrix with binary entries which are usually drawn from a Bernoulli distribution $p_{\boldsymbol{M}}$, and $\odot$ is element-wise multiplication. The objective measures how well the original sequence $\boldsymbol{X}$ can be reconstructed from a noisy version $\boldsymbol{X} \odot \boldsymbol{M}$ by the model $\text{xLSTM} : \mathbb{R}^{D \times T} \mapsto \mathbb{R}^{D \times T}$. This modeling approach, often referred to as bidirectional modeling, has proven instrumental in enabling the successful learning of protein representations at an evolutionary scale (Rives et al., 2021), which has powered many subsequent applications, such as machine-learning guided directed evolution for protein engineering (Yang et al., 2019). For details on how we extended xLSTM to the MLM setting, we refer to Appendix Section B.3.

## 3.3 RC EQUIVARIANCE

We introduce an xLSTM block equivariant to the reverse complement (RC) of an input sequence, a key property for DNA-based applications. In double-helix DNA structures, both strands are semantically equivalent, with one strand being the RC of the other. The RC strand is oriented in the opposite direction of the *forward* strand, with base pairs converted from A to T and C to G. Shrikumar et al. (2017) show that data-driven approaches to learning RC sequence equivalence can fail. To address this, Schiff et al. (2024) enforced RC-equivariance by design using two inductive biases: post-hoc conjoining (PH) (Zhou et al., 2022) and parameter sharing (PS). In PH architectures, the backbone is trained to handle both DNA sequences and their RCs by applying RC augmentations during pre-training. For downstream tasks, PH architectures are applied to both the original sequence and its RC, and their outputs are summed to reach overall RC invariance. In contrast, PS architectures integrate RC-equivariant xLSTM blocks with equivariant word embeddings and language model heads similar to Schiff et al. (2024). For additional details, see Appendix Section C.4.

## 3.4 DNA-XLSTM

For the DNA domain, we propose the DNA-xLSTM architecture to enhance sequence modeling capabilities, particularly for varying context lengths. We introduce three model configurations based on DNA-xLSTM: two sLSTM-based configurations trained with a context window of 1,024 tokens (DNA-xLSTM-500k and DNA-xLSTM-2M), and an mLSTM-based configuration trained with a context window of 32,768 tokens (DNA-xLSTM-4M). The short-context configuration, DNA-xLSTM-500k, has an embedding dimension of 128, 5 sLSTM blocks, an up-projection ratio of 1.25 to match the baseline model parameter count, and a total parameter count of 500k, while DNA-xLSTM-2M has an embedding dimension of 256, 6 sLSTM blocks, a 1.0 up-projection ratio, and 2M parameters. The long-context configuration, DNA-xLSTM-4M, has an embedding

dimension of 256, 9 mLSTM blocks, a 2.0 up-projection ratio, and is augmented with Rotary Position Encodings (RoPE) (Su et al., 2024a) to handle long-range dependencies effectively, with a total of 4M parameters. All three configurations are trained with both CLM and MLM. Furthermore, we introduce RC-equivariant versions, xLSTM-PH and xLSTM-PS, which use the original sequence and its reverse complement (RC). We trained models with these configurations on the human genome and benchmarked them against state-of-the-art DNA models, such as Transformers, DNA-Mamba (Caduceus) (Schiff et al., 2024), and HyenaDNA (Nguyen et al., 2023), showing competitive or better performance on pre-training and downstream classification tasks (Section 4.1).

## 3.5 PROT-XLSTM

For the protein domain, we propose the architectural variant Prot-xLSTM to address the complexities of protein sequence data, particularly in capturing long-range dependencies to enable homology-conditioned modeling. We introduce two configurations: Prot-xLSTM-26M and Prot-xLSTM-102M, with 26M and 102M parameters, respectively. Both configurations consist of 16 mLSTM blocks, with embedding dimensions of 512 for Prot-xLSTM-26M and 1,024 for Prot-xLSTM-102M and maintaining a consistent 2.0 up-projection ratio, and are augmented with RoPEs, across both configurations. The Prot-xLSTM models are trained using CLM with a FIM strategy on non-aligned homologous sequences with increasing context sizes ranging from 2,048 to 262,144 tokens. This enables them to perform ICL at inference time in two modes: i) generative mode, which is inherently incompatible with MLM, and ii) inpainting, which is not supported by "vanilla" CLM. Both approaches can be used for protein design, with the latter also suited for residue-based predictions, such as mutant fitness estimation. Prot-xLSTM shows better performance than similarly configured Mamba- and Transformer-based models and shows promising results for homology-conditioned sequence generation (Section 4.2).

## 3.6 CHEM-XLSTM

For the chemical sequence domain, we introduce Chem-xLSTM to enhance generative modeling of SMILES strings (Weininger, 1988), enabling domain-conditioned generation of small molecules without fine-tuning. We introduce two models: an unconditional generative model trained with a context length of 100 tokens (Chem-xLSTM-15M) and a domain-conditioned model trained with a 4,096-token context for in-context learning tasks (Chem-xLSTM-15M-icl). The latter can generate molecules within a specific domain without fine-tuning only based on examples provided as context, a highly sought-after capability in drug discovery. Both models are configured to have 15M parameters, consisting of 9 mLSTM blocks with an embedding dimension of 512 and a 1.3 up-projection ratio. The models have been benchmarked against other generative models for SMILES and at their ICL capabilities (Section 4.3).

## 4 EXPERIMENTS AND RESULTS

### 4.1 DNA SEQUENCES

For the DNA-xLSTM experiments, we followed the experimental protocol outlined in Schiff et al. (2024) and Nguyen et al. (2023) for both pre-training and downstream adaptation.

**Pre-training**. The training data for DNA-xLSTM models was sourced from the human reference genome (Church et al., 2011), which also served as the basis for our Mamba-based baseline models — HyenaDNA (Nguyen et al., 2023) and Caduceus (Schiff et al., 2024) — as well as for our Transformer++ baselines built on the Llama architecture (Touvron et al., 2023). Initially, we pretrained two variants using both CLM and MLM objectives for each: the sLSTM-based DNA-xLSTM-2M with a context length of 1,024 (Figure 2) and the mLSTM-based DNA-xLSTM-4M with a context length of 32k (Figure A3). Both models were trained with PH-equivariance (using RC augmentation without PS-equivariance). Under the CLM objective, DNA-xLSTM-2M achieved superior performance compared to Transformer++, Mamba, and HyenaDNA. This performance advantage was even more pronounced on the MLM task, where DNA-xLSTM-2M outperformed both the Transformer- and Mamba-based models.

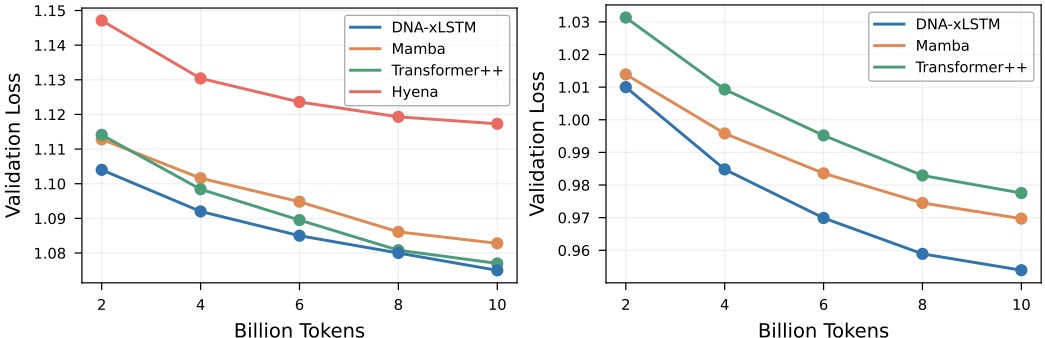

Figure 2: Pre-training of 2M-parameter DNA models on the human reference genome (GRCh38). Models are trained at single-nucleotide resolution with a context length of 1024 bases. **Left: causal language modeling.** Learning curves display **NTP loss** (↓) on a test set, plotted against the number of tokens processed. **Right: masked language modeling.** Learning curves showing **MLM loss** (↓) on a test set across the number of tokens seen for various models. In both tasks, the xLSTM-based models consistently achieve the lowest loss values across all update steps.

Similar to Caduceus, we also experimented with PS-equivariance and trained DNA-xLSTM-2M using parameter sharing for RC-equivariance under an MLM objective (Figure A4). Additionally, to match the model size used by Caduceus in one of the downstream tasks, we trained both PH- and PS-equivariant versions of DNA-xLSTM-500k with an MLM objective and a context length of 1k. These resulting PH and PS models were subsequently used for downstream fine-tuning. In Appendix Section C, we present additional pre-training results, including comparisons of large-context and PS-equivariant models. Our findings indicate that xLSTM-DNA matches or outperforms strong baselines across all pre-training settings.

**Downstream fine-tuning.** To evaluate the learned representations, we fine-tuned our MLM-pretrained, short-context DNA-xLSTM models (both PH and PS variants) on two genomic classification benchmarks. Specifically, the DNA-xLSTM-500K model was fine-tuned on the Genomic Benchmark (Grešová et al., 2023), while the DNA-xLSTM-2M model was fine-tuned on the Nucleotide Transformers Tasks (Dalla-Torre et al., 2024), which together span 18 datasets from five studies. In Table 1, we compare the performance of DNA-xLSTM-2M-PH and DNA-xLSTM-2M-PS on the Nucleotide Transformers Tasks against HyenaDNA as well as Mamba-PS and Mamba-PH. DNA-xLSTM achieved the best performance, outperforming baseline models in the sub-2M parameter range on 12 of 18 tasks and performing comparably to the much larger Nucleotide Transformer (500M parameters) by winning 8 tasks (Table A1). Additional comparisons with larger Transformer models and the performance on the Genomic Benchmark are provided in Appendix Section C.

## 4.2 PROTEIN SEQUENCES

We followed the experimental protocols from Sgarbossa et al. (2024) for protein sequences.

**Homology-aware training.** Training data was sourced from the filtered OpenProteinSet (Ahdritz et al., 2023), consisting of 270k UniClust30 clusters (508M sequences, 110B residues). Using the ProtMamba pipeline, we constructed homology-aware, alignment-free inputs by concatenating unaligned homologous sequences and mask patches for training with the FIM strategy. We trained two xLSTM-based models: Prot-xLSTM-26M and Prot-xLSTM-102M. For comparison, we also trained a smaller ProtMamba (ProtMamba-28M) and Transformer-based (Prot-Transformer++-26M) (Touvron et al., 2023) model and used the *ProtMamba Long Foundation* (ProtMamba-107M) provided by Sgarbossa et al. (2024). The initial training followed a context length scheduling strategy, with models gradually increasing context from $2^{11}$ to $2^{17}$ tokens. Finally, Prot-xLSTM-102M was further trained with $T = 2^{18}$.[2] We evaluated the models using negative log-likelihood and perplexity, calculated for different parts of the concatenated-FIM sequences. As shown in Figure 3 and Table 2, Prot-xLSTM outperformed the other architectures. Its advantage becomes even more pronounced with longer contexts, which Prot-Transformer++ cannot handle, and where Prot-xLSTM significantly

---

[2]The protein downstream tasks were evaluated using the model trained with $T$ up to $2^{17}$ tokens.

Table 1: Downstream adaption of DNA models. The performance of 2M parameter models fine-tuned on Nucleotide Transformer classification tasks on the test set is shown. PS or PH indicate models trained to be RC equivariant. Performance is averaged over 10 random seeds, and error bars indicate the difference between maximum and minimum values. The best values are highlighted in green. Bold numbers indicate statistically significant improvement over previous methods. DNA-xLSTM outperforms both Mamba and Hyena on 12 out of 18 tasks. Scores for Mamba- and Hyena-based models were obtained from Schiff et al. (2024).

| Task | Metric | HyenaDNA | Mamba-PS[a] | Mamba-PH[a] | xLSTM-PS | xLSTM-PH |
|---|---|---|---|---|---|---|
| *Histone Markers* | | | | | | |
| H3 | MCC ↑ | $0.779^{\pm0.037}$ | $0.799^{\pm0.029}$ | $0.815^{\pm0.048}$ | $0.796^{\pm0.014}$ | $\mathbf{0.824}^{\pm0.010}$ |
| H3K14AC | MCC ↑ | $0.612^{\pm0.065}$ | $0.541^{\pm0.212}$ | $0.631^{\pm0.026}$ | $0.570^{\pm0.008}$ | $0.598^{\pm0.017}$ |
| H3K36ME3 | MCC ↑ | $0.613^{\pm0.041}$ | $0.609^{\pm0.109}$ | $0.601^{\pm0.129}$ | $0.588^{\pm0.019}$ | $\mathbf{0.625}^{\pm0.010}$ |
| H3K4ME1 | MCC ↑ | $0.512^{\pm0.024}$ | $0.488^{\pm0.102}$ | $0.523^{\pm0.039}$ | $0.490^{\pm0.012}$ | $0.526^{\pm0.001}$ |
| H3K4ME2 | MCC ↑ | $0.455^{\pm0.095}$ | $0.388^{\pm0.101}$ | $0.487^{\pm0.170}$ | $\mathbf{0.489}^{\pm0.024}$ | $\mathbf{0.504}^{\pm0.012}$ |
| H3K4ME3 | MCC ↑ | $0.549^{\pm0.056}$ | $0.440^{\pm0.202}$ | $0.544^{\pm0.045}$ | $0.520^{\pm0.019}$ | $0.537^{\pm0.012}$ |
| H3K79ME3 | MCC ↑ | $0.672^{\pm0.048}$ | $0.676^{\pm0.026}$ | $0.697^{\pm0.077}$ | $0.662^{\pm0.011}$ | $0.697^{\pm0.007}$ |
| H3K9AC | MCC ↑ | $0.581^{\pm0.061}$ | $0.604^{\pm0.048}$ | $0.622^{\pm0.030}$ | $0.622^{\pm0.013}$ | $\mathbf{0.627}^{\pm0.008}$ |
| H4 | MCC ↑ | $0.763^{\pm0.044}$ | $0.789^{\pm0.020}$ | $0.811^{\pm0.022}$ | $0.793^{\pm0.011}$ | $0.813^{\pm0.008}$ |
| H4AC | MCC ↑ | $0.564^{\pm0.038}$ | $0.525^{\pm0.240}$ | $0.621^{\pm0.054}$ | $0.558^{\pm0.018}$ | $0.583^{\pm0.014}$ |
| *Regulatory Annotation* | | | | | | |
| Enhancer | MCC ↑ | $0.517^{\pm0.117}$ | $0.491^{\pm0.066}$ | $0.546^{\pm0.073}$ | $0.375^{\pm0.030}$ | $0.545^{\pm0.024}$ |
| Enhancer Types | MCC ↑ | $0.386^{\pm0.185}$ | $0.416^{\pm0.095}$ | $0.439^{\pm0.054}$ | $0.444^{\pm0.046}$ | $\mathbf{0.466}^{\pm0.011}$ |
| Promoter: All | F1 ↑ | $0.960^{\pm0.005}$ | $0.967^{\pm0.004}$ | $0.970^{\pm0.004}$ | $0.962^{\pm0.002}$ | $0.967^{\pm0.001}$ |
| NonTATA | F1 ↑ | $0.959^{\pm0.011}$ | $0.968^{\pm0.006}$ | $0.968^{\pm0.010}$ | $0.963^{\pm0.002}$ | $\mathbf{0.970}^{\pm0.001}$ |
| TATA | F1 ↑ | $0.944^{\pm0.040}$ | $0.957^{\pm0.015}$ | $0.953^{\pm0.016}$ | $0.948^{\pm0.006}$ | $0.952^{\pm0.005}$ |
| *Splice Site Annotation* | | | | | | |
| All | Accuracy ↑ | $0.956^{\pm0.011}$ | $0.927^{\pm0.021}$ | $0.940^{\pm0.027}$ | $\mathbf{0.965}^{\pm0.006}$ | $\mathbf{0.974}^{\pm0.004}$ |
| Acceptor | F1 ↑ | $0.958^{\pm0.010}$ | $0.936^{\pm0.077}$ | $0.937^{\pm0.033}$ | $\mathbf{0.970}^{\pm0.005}$ | $0.953^{\pm0.008}$ |
| Donor | F1 ↑ | $0.949^{\pm0.024}$ | $0.874^{\pm0.289}$ | $0.948^{\pm0.025}$ | $\mathbf{0.962}^{\pm0.004}$ | $0.951^{\pm0.005}$ |

[a] This method is also called Caduceus (Schiff et al., 2024).

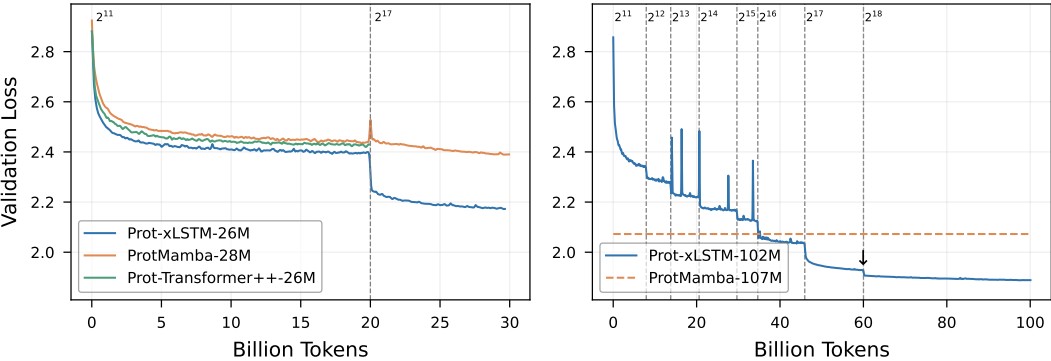

Figure 3: Generative pre-training of protein language models. The learning curves show the validation loss of homology-aware protein language models during training. **Left:** Small models trained for 20B tokens with a context size of $2^{11}$ and fine-tuned for 10B with a context of $2^{17}$ tokens. Transformer++ can only be run for a small context size. **Right:** Prot-xLSTM-102M model trained with increasing context sizes from $2^{11}$ to $2^{18}$. Vertical gray dashed lines mark the points where context size was increased. The arrow at 60B tokens indicates the model used for downstream tasks. The orange dashed line corresponds to the validation loss of ProtMamba-107M trained up to a context size of $2^{17}$ for a total of 195B tokens. Prot-xLSTM consistently outperforms other models and sets a new state-of-the-art at homology-aware generation.

outperforms ProtMamba. Furthermore, Prot-xLSTM-102M outperforms ProtMamba-107M, despite being trained on only a quarter of the total training tokens used for ProtMamba-107M. Further details are provided in the Appendix Section D.1.

Table 2: Performance comparison of protein language models at homology-conditioned generation. Test set **perplexity** ($\downarrow$) of different models with a context size of $2^{17}$ is shown across different token subsets. The average and 95% confidence interval values are computed across the test set clusters. Prot-xLSTM outperforms ProtMamba, especially when using a long context. For each token subset, the top-performing model that significantly outperforms the others, as indicated by a Wilcoxon test ($p \ll 0.05$, Figure A6), is marked bold.

| | Prot-xLSTM-26M | ProtMamba-28M | Prot-xLSTM-102M[a] | ProtMamba-107M[b] |
|---|---|---|---|---|
| All tokens | $9.41^{\pm 0.22}$ | $11.23^{\pm 0.18}$ | $\mathbf{7.35^{\pm 0.19}}$ ($7.17^{\pm 0.20}$) | $8.27^{\pm 0.18}$ |
| First seq tokens | $15.40^{\pm 0.26}$ | $15.28^{\pm 0.26}$ | $13.36^{\pm 0.35}$ ($13.42^{\pm 0.35}$) | $\mathbf{13.04^{\pm 0.36}}$ |
| Last seq tokens | $9.19^{\pm 0.30}$ | $11.08^{\pm 0.27}$ | $\mathbf{7.32^{\pm 0.29}}$ ($7.14^{\pm 0.29}$) | $8.37^{\pm 0.29}$ |
| FIM tokens | $7.23^{\pm 0.19}$ | $8.66^{\pm 0.18}$ | $\mathbf{5.87^{\pm 0.16}}$ ($5.85^{\pm 0.16}$) | $7.00^{\pm 0.16}$ |

[a] Trained for 60B tokens with $T$ up to $2^{17}$. Trained for 115B tokens with $T$ up to $2^{18}$ in parentheses.
[b] Trained for 195B tokens with $T$ up to $2^{17}$.

**Homology-conditioned protein generation.** We generated 2,500 protein sequences for each of the 19 clusters using different parameters and evaluated them across multiple metrics. The sequence similarity to natural homologs ($\min d_{\mathrm{nH}}$ and HMMER scores) and structural metrics (pLDDT and pTM) correlate with sequence perplexity, with an average absolute Spearman correlation of 0.57 across clusters for the large Prot-xLSTM model (Table A9). The distribution of these scores, such as HMMER and pLDDT, of real proteins also matches well with generated proteins. When evaluating the alignment with real-world protein score distributions, Prot-xLSTM outperforms ProtMamba across all five metrics for the small models. For the large models, Prot-xLSTM surpasses ProtMamba on three metrics (Table A11). Additional details and results can be found in Appendix Section D.2.

**Protein variant fitness prediction.** We assessed Prot-xLSTM's predictive power for mutational effects by leveraging its inpainting capability from the FIM training objective and using homologous sequences as context on the ProteinGym DMS zero-shot substitutions benchmark (Notin et al., 2023). Table A12 presents the performance comparison of Prot-xLSTM, other well-known protein models, and the current top performers on the ProteinGym leaderboard. In summary, Prot-xLSTM outperforms larger unconditional protein language models like ESM-2 (Lin et al., 2023) and ProGen2 (Nijkamp et al., 2023), and matches or surpasses ProtMamba's performance. However, it falls short compared to models that directly use the MSA, such as TranceptEVE (Notin et al., 2022b), or use structural information, like ProSST (Li et al., 2024). For further details, see Appendix Section D.3.

## 4.3 CHEMICAL SEQUENCES

**Unconditional molecule generation** aims to produce valid small organic molecules without imposing specific constraints, such as being from a particular molecular domain. Following the setup of Özçelik et al. (2024), we trained models to generate SMILES strings using a CLM approach on a dataset derived from ChEMBL with a context length of 100 tokens. We compared our Chem-xLSTM architecture with several architectures, including LSTM, GPT, S4, and Mamba, where all models contain approximately 15 million parameters. The evaluation focused on two primary metrics: perplexity and Fréchet ChemNet Distance (FCD) (Preuer et al., 2018). Chem-xLSTM achieved the lowest FCD of 0.13 and a competitive perplexity score of 1.68, indicating its strong ability to generate realistic chemical structures (see Table 3). All models produced valid, unique, and novel molecules, showcasing their effectiveness in this task. For further details and results, see Appendix Section E.1.

For **conditional molecule generation**, the objective is to generate molecules belonging to a specific molecular domain or possessing desired properties. Here, we focused on generating molecules from a particular domain using the in-context learning abilities of LLMs. To achieve this, we assembled a dataset, referred to as the *molecular domains* dataset, which comprises a diverse range of molecular domains: natural products, click-chemistry, proteolysis-targeting chimera (PROTACs), DNA-encoded chemical libraries, approved and failed drugs, and bioactive compounds from various bioassays. Molecules from the same domain are concatenated as a long sequence and augmented through permutation during training. We split the dataset into training, validation, and test domains, following an 8:1:1 ratio (Figure 4, left). The validation and test sets contained molecules from unseen domains, enabling us to evaluate the models' conditional generation capabilities through in-context learning.

Table 3: Unconditional generation of molecules with 15M parameter models. 102,400 SMILES sequences have been generated and evaluated. Error bars represent standard deviations across bootstrap resampling.Green cells highlight the best values per row. Chem-xLSTM yields the best FCD and SMILES-GPT the best perplexity. Both values are statistically significant, as indicated in bold, using a Wilcoxon test.

| | SMILES-LSTM[a] | SMILES-GPT[b] | SMILES-S4[c] | Chem-Mamba[d] | Chem-xLSTM |
|---|---|---|---|---|---|
| FCD ↓ | $0.46^{\pm <0.01}$ | $0.15^{\pm <0.01}$ | $0.28^{\pm <0.01}$ | $0.21^{\pm <0.01}$ | $\mathbf{0.13}^{\pm <0.01}$ |
| Perplexity ↓ | $1.88^{\pm 3.8}$ | $\mathbf{1.65}^{\pm 0.6}$ | $1.73^{\pm 2.4}$ | $1.74^{\pm 0.5}$ | $1.68^{\pm 1.0}$ |

[a] Segler et al. (2018)  [b] Adilov (2021)  [c] Özçelik et al. (2024)  [d] Adapted to SMILES in this work.

We trained Chem-xLSTM, Mamba, Transformer++, and S4-based models with the CLM approach on the *molecular domains* with an increased context length of 4,096 tokens. The context length for S4 models was restricted to 2,048 due to memory constraints. We evaluated the models based on NTP loss across unseen domains. The trained model Chem-xLSTM-15M-icl shows promising results in this conditional setting, outperforming the other benchmarked model classes (Figure 4, right). This demonstrates Chem-xLSTM's capability to generate molecules from an unseen chemical domain when provided with only a few exemplary molecules without fine-tuning. Further details and results are provided in Appendix Section E.2.

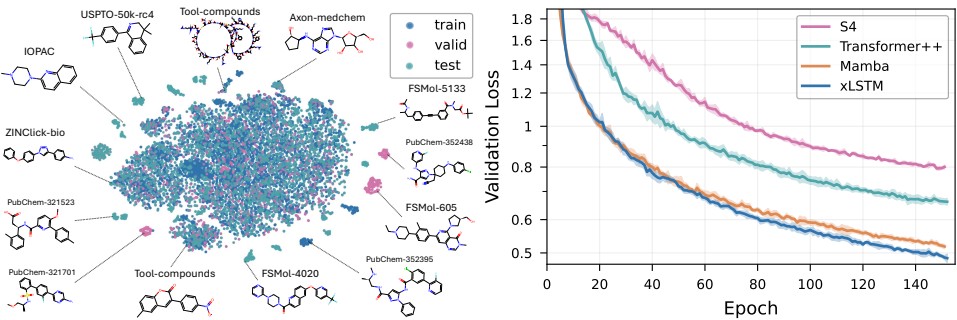

Figure 4: Conditional generation of molecules via ICL and 15M parameter models. **Left:** Visualization of different molecular domains contained in the *molecular domains* dataset. A t-SNE down-projection of molecules from different domains is shown. Clusters on the exterior represent highly specific molecular domains. The validation and test sets contain molecules from highly specific, unseen molecular domains. **Right:** Generative training of chemistry language models on the *molecular domains* dataset. Learning curves showing mean **CLM loss** (↓) on a validation set across the training epochs. Shaded areas represent the standard-deviation over runs. The Chem-xLSTM achieved the lowest loss in conditional molecule generation.

## 5 LIMITATIONS AND CONCLUSIONS

**Limitations.** Manual hyperparameter selection may not yield optimal configurations of the models, and the reliance on character-level tokenization for DNA could restrict performance with larger context sizes. Additionally, the generalizability of our models across different organisms and chemical domains is uncertain due to biases in the training datasets. Metrics like perplexity, commonly used as performance proxies, may not fully capture the true capacity of the models (Appendix G). **Conclusion.** Despite these limitations, Bio-xLSTM demonstrated effectiveness in DNA sequence modeling, performing best in masked and causal language tasks across context sizes. Our findings underpin that Bio-xLSTM is a prime candidate for foundation models in molecular biology and the models presented in this work can already be used effectively to generate DNA sequences, protein inpainting or suggesting mutations in proteins, or for conditional molecular generation without fine-tuning.

ETHICS STATEMENT

The development of our large language models for biological sequences, including DNA, proteins, and small molecules, has the potential to significantly advance biomedical research and therapeutics. In creating these models, we have taken care to train exclusively on publicly available data, such as the human reference genome, OpenProteinSet, and publicly available small molecule databases. It is important to emphasize that the models presented in this work are general base models without any specific safety tuning. These models still require human expert knowledge to produce harmful outcomes and do not possess complex instruction-following or experiment planning abilities as opposed to general-purpose language models. This significantly limits their potential for misuse by non-experts in the field. For future larger-scale model development, we recognize that additional safety measures may be necessary. One option is the exclusion of certain training samples, such as those related to human viruses, as well as the implementation of safety classifiers or guardrails that can identify and flag potentially harmful outputs. At the same time, we believe that openly sharing these base models has significant advantages for the scientific community and can aid in preparedness for addressing potential misuse scenarios. As common with machine learning methods, potential danger lies in the possibility that users rely too much on our new approach and use it without reflecting on the outcomes. However, the full pipeline, in which our method would be used, includes wet lab tests after its application, to verify and investigate the results, which decreases the danger of misuse or overly relying on the predictions. Users are encouraged to approach the model's predictions critically and consider them as one component of a broader decision-making process that incorporates expert knowledge and empirical validation.

REPRODUCIBILITY STATEMENT

To ensure reproducibility and facilitate future research, we provide three standalone code repositories for DNA-xLSTM, Prot-xLSTM, and Chem-xLSTM, each containing the respective implementations, training scripts, evaluation procedures, and pre-processed datasets.

ACKNOWLEDGEMENTS

We acknowledge EuroHPC Joint Undertaking for awarding us access to Karolina at IT4Innovations, Czech Republic; MeluXina at LuxProvide, Luxembourg; and Leonardo at CINECA, Italy.

The ELLIS Unit Linz, the LIT AI Lab, and the Institute for Machine Learning are supported by the Federal State Upper Austria. This research was partly funded by the Austrian Science Fund (FWF) [10.55776/COE12]. We thank the projects INCONTROL-RL (FFG-881064), PRI-MAL (FFG-873979), S3AI (FFG-872172), DL for GranularFlow (FFG-871302), EPILEPSIA (FFG-892171), FWF AIRI FG 9-N (10.55776/FG9), AI4GreenHeatingGrids (FFG- 899943), INTEGRATE (FFG-892418), ELISE (H2020-ICT-2019-3 ID: 951847), Stars4Waters (HORIZON-CL6-2021-CLIMATE-01-01). We thank NXAI GmbH, Audi.JKU Deep Learning Center, TGW LOGISTICS GROUP GMBH, Silicon Austria Labs (SAL), FILL Gesellschaft mbH, Anyline GmbH, Google, ZF Friedrichshafen AG, Robert Bosch GmbH, UCB Biopharma SRL, Merck Healthcare KGaA, Verbund AG, GLS (Univ. Waterloo), Software Competence Center Hagenberg GmbH, Borealis AG, TÜV Austria, Frauscher Sensonic, TRUMPF and the NVIDIA Corporation.

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

# Contents

## A    RELATED WORK

In all three areas, genomics, proteomics, and chemistry, we observe a similar trend that until around 2018 the language models were based on LSTMs (Hochreiter and Schmidhuber, 1997), then a large amount of models were based on Transformers (Vaswani et al., 2017), with different modeling paradigms and styles, and from 2023 onwards SSMs appeared.

**Language models for genomic sequence data.** DNABERT (Ji et al., 2021) and its successor DNABERT-2 (Zhou et al., 2024) are Transformer-based models that leverage bidirectional encoder representations and masked language modeling to capture nucleotide context, achieving high performance in tasks like promoter and splice site prediction. LOGO (Yang et al., 2022), another Transformer-based model, applies self-supervised learning to the human genome for sequence labeling and variant prioritization, while VIBE (Gwak and Rho, 2022) employs a hierarchical BERT model to enhance the detection of eukaryotic viruses in metagenomic data. Models like Looking-Glass (Hoarfrost et al., 2022), based on recurrent neural network (RNN), and GPN (Benegas et al., 2023), which uses convolutional neural networks (CNNs), are examples of non-Transformer-based approaches, with LookingGlass focusing on microbial genomes and GPN on plant genomes. More recent developments include the nucleotide transformer (NT) (Dalla-Torre et al., 2024), a Transformer model trained on the human genome and data from the 1000 Genomes Project, and SpeciesLM (Karollus et al., 2024), which trains Transformer-based models on 1500 fungal genomes. The latest advances, represented by Caduceus (Schiff et al., 2024) based on Mamba (Gu and Dao, 2024) and HyenaDNA (Nguyen et al., 2023), introduce SSMs that allow generative modeling and representation learning for long DNA sequences.

**Language models for protein sequence data.** Until around 2019, the field was dominated by RNNs and LSTM-based models trained with CLM. Notable examples include UniRep (Alley et al., 2019), which employed multiplicative-LSTM to capture rich protein representations, and SSA (Bepler and Berger, 2019), which used bidirectional RNNs for structural similarity prediction. Since then the field has shifted towards Transformer-based models and MLM, driven by their success in natural language processing. Early adopters of this shift included the TAPE benchmark for protein downstream tasks (Rao et al., 2019), which evaluated both an LSTM and a Transformer architecture trained with CLM and MLM, respectively. Elnaggar et al. (2021) further expanded the use of Transformers with large-scale MLM, setting new benchmarks in protein sequence analysis with Prot-T5. ESM (Rives et al., 2021) applied MLM to a Transformer on a massive scale, capturing evolutionary patterns across diverse protein sequences. Other significant Transformer-based models include MSA-Transformer (Rao et al., 2021), which applied MLM to multiple-sequence alignments (MSA), and ProGen (Madani et al., 2023), which used CLM and Transformers for controlled protein sequence generation. Additionally, models like ProtGPT2 (Ferruz et al., 2022) and ProteinBERT (Brandes et al., 2022) utilized the power of Transformer architectures in generating novel protein sequences and functional predictions. Furthermore, (Su et al., 2024b) introduced a "structure-aware vocabulary" which they use as input for Transformer-based models. The recently proposed PoET (Truong Jr and Bepler, 2023) is an autoregressive Transformer model trained on non-aligned homologous sequences, providing a novel approach for conditional protein design and protein fitness prediction. Building on the concept of non-aligned homologous sequences, ProtMamba (Sgarbossa et al., 2024) leverages emerging SSMs to manage long-context conditioning on proteins, effectively utilizing autoregressive and FIM strategies. For a more comprehensive review of these advancements, including their applications in functional protein design, see Notin et al. (2024) and Hu et al. (2022).

**Language models for chemical sequence data.** The first language model for chemical sequences was an LSTM-based, autoregressive method developed by Segler et al. (2018), which demonstrated that the SMILES syntax (Weininger, 1988) and generation of realistic organic molecules can be learned. Honda et al. (2019) introduced a Transformer model for this task, showing that this leads to informative representations of molecules. The Molecular Transformer (Schwaller et al., 2019) consists of a Transformer-based encoder and decoder, trained on chemical reaction data to translate between reactants and products. More recently, SSMs have been used for generative modeling of SMILES strings (Özçelik et al., 2024). Subsequent models such as MolGPT (Bagal et al., 2021) and cMolGPT (Wang et al., 2023) utilized the GPT architecture to generate SMILES strings, with MolGPT conditioning on chemical properties and scaffolds, and cMolGPT focusing on biomolecular targets. Transformer-based approaches have also been employed to optimize the properties of small molecules in a reinforcement-learning setting (Mazuz et al., 2023). Encoder-style language models

for chemistry, such as SmilesLSTM (Mayr et al., 2018), ChemNet (Preuer et al., 2018), and CNN-based models (Jastrzębski et al., 2016), initially used activity and property prediction as pre-training or training objectives. Later, these encoder-style language models were trained with the masking language modeling objective, as seen in ChemBERTA (Chithrananda et al., 2020), Chemberta-2 (Ahmad et al., 2022), SMILES-BERT (Wang et al., 2019b), MolFormer (Ross et al., 2022) and rxnfp-BERT (Schwaller et al., 2021). Some models have also adopted contrastive objectives (Seidl et al., 2023). Large language models for molecules have also been shown to learn complex molecular distributions (Flam-Shepherd et al., 2022). For a more thorough and comprehensive overview, we refer to the surveys by Bran and Schwaller (2024) and Zhang et al. (2025)

## B xLSTM ARCHITECTURE DETAILS

### B.1 xLSTM AND BIO-xLSTM BLOCKS

Beck et al. (2024) suggested xLSTM blocks, which are residual (Srivastava et al., 2015; He et al., 2016) block modules, into which the sLSTM and mLSTM layers can be integrated. The two basic blocks can in principle be characterized by either applying post-sLSTM/mLSTM up- and down-projections (similar to Vaswani et al. (2017)) or by applying pre-sLSTM/mLSTM up-projections and post-sLSTM/mLSTM down-projections (similar to Dao (2024)). An sLSTM block integrates the sLSTM layer into the up- and down-projection block, while the mLSTM block integrates the mLSTM layer into the pre-up-projection and post-down-projection block. The two basic xLSTM blocks also make use of neural network modules like layer normalization (Ba et al., 2016), short causal convolutions, and, group normalization (Wu and He, 2020). For the exact architecture of the blocks, we refer to Beck et al. (2024, Sec.2.4). An xLSTM architecture is constructed by residually stacking the suggested xLSTM blocks. For that, the most commonly used pre-LayerNorm residual backbone is used.

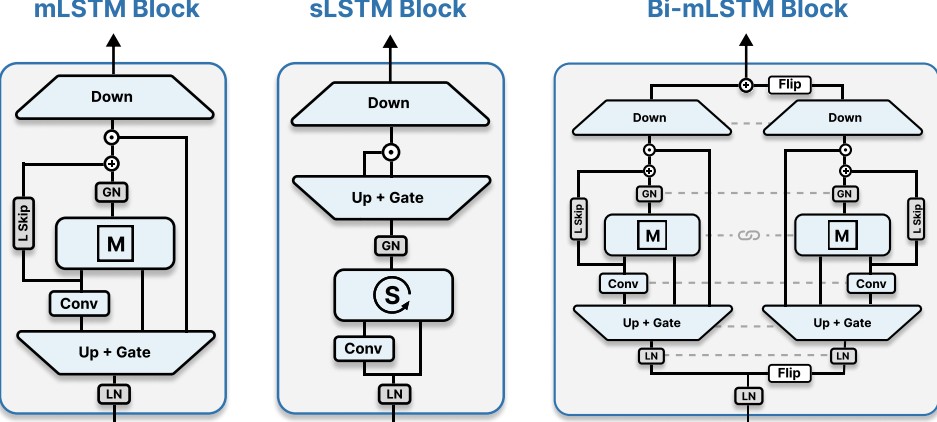

Figure A1: xLSTM and Bio-xLSTM blocks. **Left: mLSTM block.** *LN* (Layer Normalization) and *GN* (Group Normalization) refer to normalization modules, while *L Skip* represents learnable skip connections and *Conv* denotes causal 1D convolutions. The mLSTM block utilizes a gated pre-up-projection structure, akin to modern State-Space Models, with gates activated by the Swish function. **Middle: sLSTM block.** The sLSTM block features a GELU-gated post-up-projection structure, similar to Transformer architectures. **Right: Bidirectional mLSTM block.** For bidirectional processing, the xLSTM applies each block to the input sequence twice before combining the outputs: once left-to-right and once right-to-left.

For Bio-xLSTM we keep the basic xLSTM building blocks and the basic xLSTM architecture template, but adjust them to the respective domains. Figure A1 depicts sLSTM and mLSTM blocks, as well as, a bidirectional mLSTM configuration with weight-tied layers.

### B.2 MODES OF OPERATION: PARALLEL, CHUNKWISE OR RECURRENT

Similar to linear attention variants (Katharopoulos et al., 2020; Yang et al., 2024), the mLSTM has three possible formulations: parallel, recurrent or chunkwise. The presentation in section 2.2 (and Beck et al., 2024) focuses on the recurrent form:

$$\boldsymbol{C}_t = \sigma(\tilde{f}_t)\,\boldsymbol{C}_{t-1} + \exp(\tilde{i}_t) \odot \boldsymbol{v}_t \boldsymbol{k}_t^\mathsf{T}$$

$$\boldsymbol{n}_t = \sigma(\tilde{f}_t)\,\boldsymbol{n}_{t-1} + \exp(\tilde{i}_t)\,\boldsymbol{k}_t$$

$$\boldsymbol{h}_t = \sigma(\tilde{\boldsymbol{o}}_t) \odot \frac{\boldsymbol{C}_t \boldsymbol{q}_t}{\max(|\boldsymbol{n}_t \boldsymbol{q}_t|, 1)}.$$

This form is especially useful for inference when samples arrive one time-step at a time.

The omission of the recurrent connections in mLSTM allows for a parallel formulation (Beck et al., 2024, Appendix):

$$\tilde{\boldsymbol{F}} = \begin{bmatrix} 0 & 0 & 0 & \cdots & 0 \\ \ln\sigma(\tilde{f}_2) & 0 & 0 & \cdots & 0 \\ \ln\sigma(\tilde{f}_2) + \ln\sigma(\tilde{f}_3) & \ln\sigma(\tilde{f}_3) & 0 & \cdots & 0 \\ \vdots & \vdots & \vdots & \ddots & \vdots \\ \sum_{t=2}^{T}\ln\sigma(\tilde{f}_t) & \sum_{t=3}^{T}\ln\sigma(\tilde{f}_t) & \sum_{t=4}^{T}\ln\sigma(\tilde{f}_t) & \cdots & 0 \end{bmatrix}$$

$$\boldsymbol{D} = \exp\big(\tilde{\boldsymbol{F}} + \boldsymbol{1} \otimes \tilde{\boldsymbol{i}}\big) \odot \boldsymbol{M} \tag{19}$$

$$\boldsymbol{H} = \sigma(\tilde{\boldsymbol{O}}) \odot \frac{\boldsymbol{D} \odot \boldsymbol{Q}\boldsymbol{K}^{\mathsf{T}}}{\max\big(|\boldsymbol{D} \odot \boldsymbol{Q}\boldsymbol{K}^{\mathsf{T}}| \cdot \boldsymbol{1}, \boldsymbol{1}\big)}\boldsymbol{V},$$

where $\boldsymbol{Q}, \boldsymbol{K}, \boldsymbol{V}, \tilde{\boldsymbol{O}} \in \mathbb{R}^{T \times d}, \tilde{\boldsymbol{i}} \in \mathbb{R}^T$ and $\boldsymbol{M} \in \{0, 1\}^{T \times T}$ is a causal (i.e. lower-triangular) masking matrix. The $\otimes$ refers to an outer product, while $\odot$ is a Hadamard (i.e. element-wise) product. The fraction, max and other non-linear functions are also applied element-wise. This parallel form enables an efficient training regime, similar to Transformers.

The chunkwise formulation is a hybrid of the recurrent and parallel forms:

$$\tilde{\boldsymbol{F}} = \begin{bmatrix} 0 & 0 & 0 & \cdots & 0 \\ \ln\sigma(\tilde{f}_{t-C+2}) & 0 & 0 & \cdots & 0 \\ \ln\sigma(\tilde{f}_{t-C+2}) + \ln\sigma(\tilde{f}_{t-C+3}) & \ln\sigma(\tilde{f}_{t-C+3}) & 0 & \cdots & 0 \\ \vdots & \vdots & \vdots & \ddots & \vdots \\ \sum_{\tau=2}^{C}\ln\sigma(\tilde{f}_{t-C+\tau}) & \sum_{\tau=3}^{C}\ln\sigma(\tilde{f}_{t-C+\tau}) & \sum_{\tau=4}^{C}\ln\sigma(\tilde{f}_{t-C+\tau}) & \cdots & 0 \end{bmatrix}$$

$$\boldsymbol{D} = \exp\big(\tilde{\boldsymbol{F}} + \boldsymbol{1} \otimes \tilde{\boldsymbol{i}}\big) \odot \boldsymbol{M}$$

$$\boldsymbol{f} = \left(\sigma(\tilde{f}_{t-C+1}), \sigma(\tilde{f}_{t-C+1})\,\sigma(\tilde{f}_{t-C+2}), \ldots, \prod_{\tau=1}^{C}\sigma(\tilde{f}_{t-C+\tau})\right)$$

$$\boldsymbol{H} = \sigma(\tilde{\boldsymbol{O}}) \odot \frac{\big(\boldsymbol{D} \odot \boldsymbol{Q}\boldsymbol{K}^{\mathsf{T}}\big)\boldsymbol{V} + \operatorname{diag}(\boldsymbol{f})\,\boldsymbol{Q}\boldsymbol{C}_{t-C}^{\mathsf{T}}}{\max\big(|(\boldsymbol{D} \odot \boldsymbol{Q}\boldsymbol{K}^{\mathsf{T}}) \cdot \boldsymbol{1} + \operatorname{diag}(\boldsymbol{f})\,\boldsymbol{Q}\boldsymbol{n}_{t-C}|, \boldsymbol{1}\big)}$$

$$\boldsymbol{C}_t = \left(\prod_{\tau=1}^{C}\sigma(\tilde{f}_{t-C+\tau})\right)\boldsymbol{C}_{t-C} + \boldsymbol{V}^{\mathsf{T}}\operatorname{diag}(\boldsymbol{d}_C)\,\boldsymbol{K}$$

$$\boldsymbol{n}_t = \left(\prod_{\tau=1}^{C}\sigma(\tilde{f}_{t-C+\tau})\right)\boldsymbol{n}_{t-C} + \boldsymbol{K}^{\mathsf{T}}\boldsymbol{d}_C,$$

with $\boldsymbol{Q}, \boldsymbol{K}, \boldsymbol{V}, \tilde{\boldsymbol{O}} \in \mathbb{R}^{C \times d}$ and $\tilde{\boldsymbol{i}} \in \mathbb{R}^C$ the pre-activations from $t - C + 1$ to $t$. Furthermore, $\boldsymbol{M} \in \{0, 1\}^{C \times C}$, is a local causal (i.e. lower-triangular) masking matrix, $\boldsymbol{d}_C$ denotes the last row of $\boldsymbol{D}$, diag transforms a vector into a diagonal matrix, and $C$ is the chunk size. The chunkwise formulation allows for implementing hardware-aware efficient kernels (Beck et al., 2025). For $C = 1$, we recover the recurrent form, whereas for $C = T$, we obtain the parallel form.

## B.3 EFFICIENT BIDIRECTIONAL MODELING FOR WEIGHT-TIED LAYERS OF BIO-XLSTM

Bidirectional modeling is often required to learn representations of biological and chemical sequences, for example with the MLM paradigm. The default approach for bidirectional modeling would be to use an mLSTM layer on the usual sequence $\boldsymbol{X}_{1:T} = (\boldsymbol{x}_1, \boldsymbol{x}_2, \ldots, \boldsymbol{x}_T)$ and then applying a weight-tied layer on the reversed sequence $\boldsymbol{X}_{T:1} = (\boldsymbol{x}_T, \boldsymbol{x}_{T-1}, \ldots, \boldsymbol{x}_1)$ and subsequently summing those outputs:

$$\boldsymbol{H}^+ = \mathrm{mLSTM}(\boldsymbol{X}_{1:T}; \boldsymbol{w}) \tag{20}$$

$$\boldsymbol{H}^- = \mathrm{mLSTM}(\boldsymbol{X}_{T:1}; \boldsymbol{w}) \tag{21}$$

$$\boldsymbol{H} = \boldsymbol{H}^+ + \boldsymbol{H}_{T:1}^-, \tag{22}$$

where $\boldsymbol{H}_{T:1}^{-}$ indicates that the sequence is reversed again, and $\boldsymbol{w}$ are the parameters of the LSTM-layer $\mathrm{mLSTM}(\boldsymbol{X}_{1:T}; \boldsymbol{w})$ which are assumed to be the same for both directions, i.e. weight-tied. This approach is schematically depicted in Figure A1 (Right). However, this approach is inefficient with respect to memory and operations because it has to calculate and store all internal quantities, such as the cell states, twice for the backward pass. A variant of this approach is to perform the forward direction in one block (Eq. 20) and the reverse direction in a consecutive block (Eq. 21) of the architecture (Alkin et al., 2024).

**We propose an efficient bidirectional modeling approach.** Instead of running the network twice, it is possible to compute both $\boldsymbol{H}^{+}$ and $\boldsymbol{H}_{T:1}^{-}$, with a single forward pass. This effectively reduces computational demands and memory. The simultaneous computation is especially interesting for the parallel formulation of the mLSTM. Concretely, $\boldsymbol{H}^{+}$ can directly be computed using equation (19) and

$$\boldsymbol{H}_{T:1}^{-} = \sigma(\tilde{\boldsymbol{O}}_{T:1}^{-}) \odot \frac{\boldsymbol{D}_{T:1,1:T}^{-} \odot \boldsymbol{Q}_{T:1}^{-} \boldsymbol{K}^{-\mathsf{T}}}{\max\big(|\boldsymbol{D}_{T:1,1:T}^{-} \odot \boldsymbol{Q}_{T:1}^{-} \boldsymbol{K}^{-\mathsf{T}}| \cdot \mathbf{1}, \mathbf{1}\big)} \boldsymbol{V}^{-}$$

$$= \sigma(\tilde{\boldsymbol{O}}^{+}) \odot \frac{\boldsymbol{D}_{T:1,T:1}^{-} \odot \boldsymbol{Q}^{+} \boldsymbol{K}^{+\mathsf{T}}}{\max\big(|\boldsymbol{D}_{T:1,T:1}^{-} \odot \boldsymbol{Q}_{T:1}^{-} \boldsymbol{K}^{+\mathsf{T}}| \cdot \mathbf{1}, \mathbf{1}\big)} \boldsymbol{V}^{+}.$$

Because $\boldsymbol{D}^{+}$ is upper triangular, $\boldsymbol{D}_{T:1,T:1}^{-}$ is lower triangular and the sum can be written as:

$$\boldsymbol{H}^{+} + \boldsymbol{H}_{T:1}^{-} = \sigma(\tilde{\boldsymbol{O}}^{+}) \odot \frac{\boldsymbol{D} \odot \boldsymbol{Q}^{+} \boldsymbol{K}^{+\mathsf{T}}}{\max\big(|\boldsymbol{D} \odot \boldsymbol{Q}_{T:1}^{-} \boldsymbol{K}^{+\mathsf{T}}| \cdot \mathbf{1}, \mathbf{1}\big)} \boldsymbol{V}^{+},$$

where

$$\boldsymbol{D} = \boldsymbol{D}^{+} + \boldsymbol{D}_{T:1,T:1}^{-} = \exp(\tilde{\boldsymbol{F}} + \mathbf{1} \otimes \tilde{\boldsymbol{i}})$$

$$\tilde{\boldsymbol{F}} = \begin{bmatrix} 0 & \ln\sigma(\tilde{f}_1) & \cdots & \sum_{t=1}^{T-2}\ln\sigma(\tilde{f}_t) & \sum_{t=1}^{T-1}\ln\sigma(\tilde{f}_t) \\ \ln\sigma(\tilde{f}_2) & 0 & & \vdots & \vdots \\ \vdots & & 0 & \ln\sigma(\tilde{f}_{T-2}) & \ln\sigma(\tilde{f}_{T-2})+\ln\sigma(\tilde{f}_{T-1}) \\ \sum_{t=2}^{T-1}\ln\sigma(\tilde{f}_t) & \cdots & \ln\sigma(\tilde{f}_{T-1}) & 0 & \ln\sigma(\tilde{f}_{T-1}) \\ \sum_{t=2}^{T}\ln\sigma(\tilde{f}_t) & \sum_{t=3}^{T}\ln\sigma(\tilde{f}_t) & \cdots & \ln\sigma(\tilde{f}_T) & 0 \end{bmatrix}.$$

Note that this principle can also be applied in the parallel part of the chunk-wise formulation (see B.2).

## C DNA-xLSTM: DETAILS AND ADDITIONAL RESULTS

In this section, we provide further details regarding the architecture, training setup, and evaluation metrics for the DNA-xLSTM models.

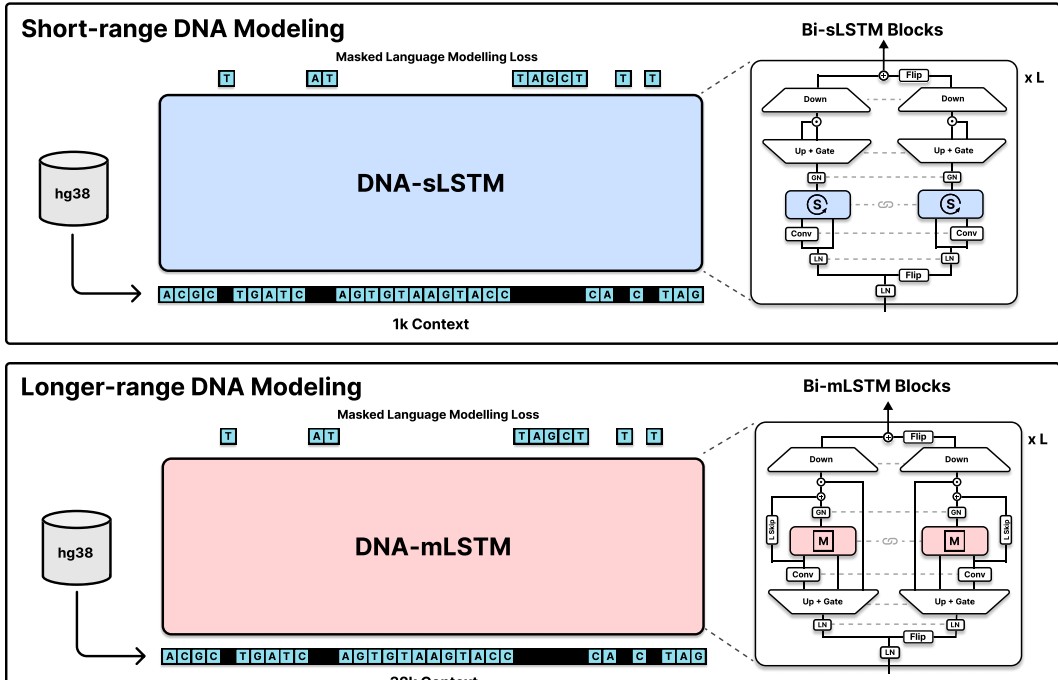

Figure A2: **DNA-xLSTM Architecture**. The top panel shows DNA-sLSTM, trained with masked language modeling for short-range contexts (1,024 nucleotides) using Bi-sLSTM blocks. The bottom panel depicts DNA-mLSTM, designed for long-range contexts (32,768 nucleotides) with sequence-parallel mLSTM blocks. Both models are trained on the human reference genome (hg38).

### C.1 PRE-TRAINING

**Experimental setup.** We followed the experimental protocol established in Schiff et al. (2024) and Nguyen et al. (2023). The human reference genome (Church et al., 2011) was used as the training dataset for two main tasks: **a)** causal language modeling (CLM) and **b)** masked language modeling (MLM). We employed context lengths of 1,024 and 32,000 tokens for these tasks.

To ensure a fair comparison with previous methods, such as Schiff et al. (2024), we used character- or base pair-level tokenization, training models with parameter sizes ranging from 500k to 4M. This experimental setup enabled us to evaluate model performance for both **a)** generative modeling of DNA sequences and **b)** learning rich DNA sequence representations—core tasks in this domain.

**Methods and hyperparameters.** In our pre-training experiments, we compared several architectures: a Transformer variant based on the Llama architecture, referred to as Transformer++ (Touvron et al., 2023), DNA-xLSTM, HyenaDNA (Nguyen et al., 2023), and DNA-Mamba (also known as Caduceus) (Schiff et al., 2024). Each architecture was trained under both CLM and MLM settings. Additionally, we assessed two types of reverse-complement (RC) equivariant models when applicable: DNA-Mamba-PH and DNA-Mamba-PS, as well as DNA-xLSTM-PH and DNA-xLSTM-PS. For non-equivariant models, reverse-complement augmentation was applied, following the approach described in Schiff et al. (2024). Further details on RC-equivariant modeling can be found in Section C.4. The hyperparameters for DNA-xLSTM and Transformer++ were optimized using a validation set, with the final configurations reported in Appendix Tables A4 and A3.

**Metrics.** We report cross-entropy loss on a held-out test set for both CLM and MLM pre-training experiments.

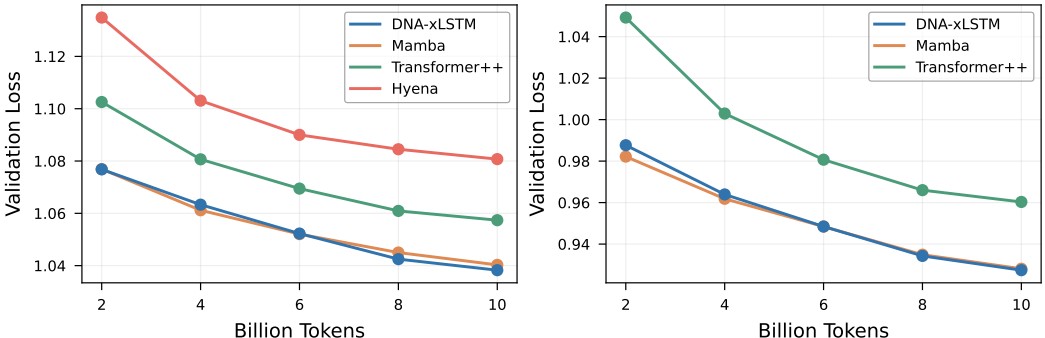

Figure A3: Pre-training of 4M-parameter DNA models on the human reference genome (GRCh38). The models are trained on the human reference genome at single-nucleotide resolution with a context length of 32k bases. **Left: causal language modeling.** Learning curves display **CLM loss** (↓) on a held-out test set, plotted against the number of tokens processed. **Right: masked language modeling.** Learning curves for bidirectional models trained with the **MLM** objective (↓). The DNA-xLSTM-4M model outperforms both Transformer++ and Hyena-DNA models of similar size, and matches the performance of Caduceus-4M.

**Results.** Our experiments show that the sLSTM-based DNA-xLSTM-2M model, trained with a context size of 1,024 and reverse-complement augmentation, outperforms DNA-Mamba (Schiff et al., 2024), HyenaDNA (Nguyen et al., 2023), and Transformer++ across both CLM and MLM tasks. Notably, the performance gap between DNA-xLSTM and the baseline models increases in the MLM setting. See Figure 2.

We further enhanced DNA-xLSTM-500k and DNA-xLSTM-2M models by incorporating reverse-complement equivariance via parameter sharing. For smaller models, we achieved MLM losses comparable to DNA-Mamba-PS, with a significant improvement over DNA-Mamba-PS as model size scaled to 2M parameters (Figure A4). Additionally, we pre-trained a long-range DNA-xLSTM model based on mLSTM, with a context size of 32k, using both CLM and MLM objectives. This model achieved the lowest cross-entropy loss in both tasks, outperforming Transformers and HyenaDNA, while performing comparably to Mamba (Figure A3).

## C.2 DOWNSTREAM TASKS

**Experimental setup.** Two sets of downstream tasks were used for evaluating the learned representations: the Genomic benchmark (Grešová et al., 2023) and the Nucleotide Transformers Tasks (Dalla-Torre et al., 2024), which is a collection of 18 datasets derived from five peer-reviewed studies (Phaml et al., 2005; Oubounyt et al., 2019; Wang et al., 2019a; Scalzitti et al., 2021; Geng et al., 2022). These classification tasks were selected to determine how rich the learned representations of the architectures are. To extract representations from the pre-trained xLSTM-DNA models, we perform average pooling on the activations from the final xLSTM block. For each downstream dataset, these representations served as inputs to a task-specific classification head that were jointly fine-tuned with the pre-trained model parameters.

**Methods and hyperparameters.** For Nucleaotide Transformer tasks, we compared HyenaDNA, DNA-Mamba, and xLSTM-based models pre-trained with 2M parameters. For Genomic benchmark tasks, we compare the smaller xLSTM-500k against Mamba. In both settings, models were pre-trained with a context size of 1,024.

**Metrics.** For the Nucleotide Transformer downstream tasks different metrics are used depending on the type of task: MCC was used for histone markers and enhancer annotation, F1-score was used for promoter annotation and splice site acceptor/donor, and accuracy was used for the splice site. The downstream performance on the Genomic benchmark was evaluated using the Top-1 accuracy.

**Results.** On the extensive set of downstream tasks, DNA-xLSTM is the best model with fewer than 2M parameters outperforming other small models on 12 of 18 tasks. In a comparison with much larger models, DNA-xLSTM and is on par with the 500M parameter model Nucleotide Transformer

Table A1: Downstream adaption of DNA models (extended version). The test set performance of DNA models with 2M parameters and models with over 100M parameters, fine-tuned on Nucleotide Transformer classification tasks, is shown. Models marked with PS or PH are trained to be RC equivariant. The used metric is provided in the *Metric* column and best values are highlighted in green Results are averaged over 10 random seeds, with error bars representing the difference between the maximum and minimum values across the runs. The best scores are highlighted in green. xLSTM-DNA-PH with 2M parameters outperforms similarly sized Hyena- and Mamba-based models, while achieving comparable results to the much larger Nucleotide Transformer. Scores for all models except xLSTM were obtained from Schiff et al. (2024).

| Task | Metric | > 100M Param. Models | | | 2M Param. Models | | | | |
|---|---|---|---|---|---|---|---|---|---|
| | | Enformer (252M) | DNABERT-2 (117M) | NT-v2 (500M) | HyenaDNA | Mamba-PS | Mamba-PH | xLSTM-PS | xLSTM-PH |
| *Histone Markers* | | | | | | | | | |
| H3 | MCC ↑ | $0.719^{\pm0.048}$ | $0.785^{\pm0.033}$ | $0.784^{\pm0.047}$ | $0.779^{\pm0.037}$ | $0.799^{\pm0.029}$ | $0.815^{\pm0.048}$ | $0.796^{\pm0.014}$ | $0.824^{\pm0.010}$ |
| H3K14AC | MCC ↑ | $0.288^{\pm0.077}$ | $0.516^{\pm0.028}$ | $0.551^{\pm0.021}$ | $0.612^{\pm0.065}$ | $0.541^{\pm0.212}$ | $0.631^{\pm0.026}$ | $0.570^{\pm0.008}$ | $0.598^{\pm0.017}$ |
| H3K36ME3 | MCC ↑ | $0.344^{\pm0.055}$ | $0.591^{\pm0.020}$ | $0.625^{\pm0.030}$ | $0.613^{\pm0.041}$ | $0.609^{\pm0.109}$ | $0.601^{\pm0.129}$ | $0.588^{\pm0.019}$ | $0.625^{\pm0.010}$ |
| H3K4ME1 | MCC ↑ | $0.291^{\pm0.061}$ | $0.511^{\pm0.028}$ | $0.550^{\pm0.021}$ | $0.512^{\pm0.024}$ | $0.488^{\pm0.102}$ | $0.523^{\pm0.039}$ | $0.490^{\pm0.012}$ | $0.526^{\pm0.001}$ |
| H3K4ME2 | MCC ↑ | $0.211^{\pm0.069}$ | $0.336^{\pm0.040}$ | $0.319^{\pm0.045}$ | $0.455^{\pm0.095}$ | $0.388^{\pm0.101}$ | $0.487^{\pm0.170}$ | $0.489^{\pm0.024}$ | $0.504^{\pm0.012}$ |
| H3K4ME3 | MCC ↑ | $0.158^{\pm0.072}$ | $0.352^{\pm0.077}$ | $0.410^{\pm0.033}$ | $0.549^{\pm0.056}$ | $0.440^{\pm0.202}$ | $0.544^{\pm0.045}$ | $0.520^{\pm0.019}$ | $0.537^{\pm0.012}$ |
| H3K79ME3 | MCC ↑ | $0.496^{\pm0.042}$ | $0.613^{\pm0.030}$ | $0.626^{\pm0.046}$ | $0.672^{\pm0.048}$ | $0.676^{\pm0.026}$ | $0.697^{\pm0.077}$ | $0.662^{\pm0.011}$ | $0.697^{\pm0.007}$ |
| H3K9AC | MCC ↑ | $0.420^{\pm0.063}$ | $0.542^{\pm0.029}$ | $0.562^{\pm0.040}$ | $0.581^{\pm0.061}$ | $0.604^{\pm0.048}$ | $0.622^{\pm0.030}$ | $0.622^{\pm0.013}$ | $0.627^{\pm0.008}$ |
| H4 | MCC ↑ | $0.732^{\pm0.076}$ | $0.796^{\pm0.027}$ | $0.799^{\pm0.025}$ | $0.763^{\pm0.044}$ | $0.789^{\pm0.020}$ | $0.811^{\pm0.022}$ | $0.793^{\pm0.011}$ | $0.813^{\pm0.008}$ |
| H4AC | MCC ↑ | $0.273^{\pm0.063}$ | $0.463^{\pm0.041}$ | $0.495^{\pm0.032}$ | $0.564^{\pm0.038}$ | $0.525^{\pm0.240}$ | $0.621^{\pm0.054}$ | $0.558^{\pm0.018}$ | $0.583^{\pm0.014}$ |
| *Regulatory Annotation* | | | | | | | | | |
| Enhancer | MCC ↑ | $0.451^{\pm0.108}$ | $0.516^{\pm0.098}$ | $0.548^{\pm0.144}$ | $0.517^{\pm0.117}$ | $0.491^{\pm0.066}$ | $0.546^{\pm0.073}$ | $0.375^{\pm0.030}$ | $0.545^{\pm0.024}$ |
| Enhancer Types | MCC ↑ | $0.309^{\pm0.134}$ | $0.423^{\pm0.051}$ | $0.424^{\pm0.132}$ | $0.386^{\pm0.185}$ | $0.416^{\pm0.095}$ | $0.439^{\pm0.054}$ | $0.444^{\pm0.046}$ | $0.466^{\pm0.011}$ |
| Promoter: All | F1 ↑ | $0.954^{\pm0.006}$ | $0.971^{\pm0.006}$ | $0.976^{\pm0.006}$ | $0.960^{\pm0.005}$ | $0.967^{\pm0.004}$ | $0.970^{\pm0.004}$ | $0.962^{\pm0.002}$ | $0.967^{\pm0.001}$ |
| NonTATA | F1 ↑ | $0.955^{\pm0.010}$ | $0.972^{\pm0.005}$ | $0.976^{\pm0.006}$ | $0.959^{\pm0.011}$ | $0.968^{\pm0.006}$ | $0.968^{\pm0.010}$ | $0.963^{\pm0.002}$ | $0.970^{\pm0.001}$ |
| TATA | F1 ↑ | $0.960^{\pm0.023}$ | $0.955^{\pm0.021}$ | $0.966^{\pm0.013}$ | $0.944^{\pm0.040}$ | $0.957^{\pm0.015}$ | $0.953^{\pm0.016}$ | $0.948^{\pm0.006}$ | $0.952^{\pm0.005}$ |
| *Splice Site Annotation* | | | | | | | | | |
| All | Accuracy ↑ | $0.848^{\pm0.019}$ | $0.939^{\pm0.009}$ | $0.983^{\pm0.008}$ | $0.956^{\pm0.011}$ | $0.927^{\pm0.021}$ | $0.940^{\pm0.027}$ | $0.965^{\pm0.006}$ | $0.974^{\pm0.004}$ |
| Acceptor | F1 ↑ | $0.914^{\pm0.028}$ | $0.975^{\pm0.006}$ | $0.981^{\pm0.011}$ | $0.958^{\pm0.010}$ | $0.936^{\pm0.077}$ | $0.937^{\pm0.033}$ | $0.970^{\pm0.005}$ | $0.953^{\pm0.008}$ |
| Donor | F1 ↑ | $0.906^{\pm0.027}$ | $0.963^{\pm0.006}$ | $0.985^{\pm0.022}$ | $0.949^{\pm0.024}$ | $0.948^{\pm0.025}$ | $0.874^{\pm0.289}$ | $0.962^{\pm0.004}$ | $0.951^{\pm0.005}$ |

Table A2: Downstream adaption of DNA language models on the Genomics Benchmarks. The Top-1 **accuracy** (↑) for RC-equivariant PS and PH xLSTM and Mamba-based Caduceus models, both with 500k parameters, are shown. Error bars represent the range of scores across five random seeds. xLSTM achieves comparable overall performance to Mamba and demonstrates superior accuracy when both models employ post-hoc conjoining. Scores for all models except xLSTM were obtained from Schiff et al. (2024).

| | Mamba-PH-500k | xLSTM-PH-500k ‖ | Mamba-PS-500k | xLSTM-PS-500k |
|---|---|---|---|---|
| Mouse Enhancers | $0.754^{\pm0.074}$ | $0.780^{\pm0.018}$ | $0.793^{\pm0.058}$ | $0.778^{\pm0.007}$ |
| Coding. vs. Intergenomic | $0.915^{\pm0.003}$ | $0.931^{\pm0.001}$ | $0.910^{\pm0.003}$ | $0.934^{\pm0.002}$ |
| Human vs. Worm | $0.973^{\pm0.001}$ | $0.965^{\pm0.001}$ | $0.968^{\pm0.002}$ | $0.956^{\pm0.001}$ |
| Human Enhancers Cohn | $0.747^{\pm0.004}$ | $0.742^{\pm0.005}$ | $0.745^{\pm0.007}$ | $0.734^{\pm0.005}$ |
| Human Enhancers Ensemble | $0.893^{\pm0.008}$ | $0.920^{\pm0.001}$ | $0.900^{\pm0.006}$ | $0.902^{\pm0.004}$ |
| Human Regulatory | $0.872^{\pm0.011}$ | $0.872^{\pm0.002}$ | $0.873^{\pm0.007}$ | $0.869^{\pm0.005}$ |
| Human OCR Ensembl | $0.828^{\pm0.006}$ | $0.826^{\pm0.002}$ | $0.818^{\pm0.006}$ | $0.800^{\pm0.002}$ |
| Human NonTATA Promoters | $0.946^{\pm0.007}$ | $0.951^{\pm0.004}$ | $0.945^{\pm0.010}$ | $0.949^{\pm0.001}$ |

(NT-v2) winning 8 of 18 tasks (see Table A1). On the Genomic benchmark, DNA-xLSTM is overall on par with Mamba-DNA and shows especially strong results with posthoc conjoining, winning 5 of 8 tasks compared to Mamba-DNA-PH. Results are reported in Table A2.

## C.3 ARCHITECTURE AND HYPERPARAMETERS

The hyperparameters and composition of the DNA-xLSTM and DNA-Transformer++ models for pre-training with context size 1k and 32k are reported in Tables A4 and A3. The hyperparameters were selected on a separate validation set using manual hyperparameter selection due to limited computational resources.

## C.4 REVERSE-COMPLEMENT INVARIANCE

We develop an xLSTM version that is invariant to the RC of an input sequence which is relevant for DNA applications following Schiff et al. (2024). In double-helix DNA structures, both strands

Table A3: Pre-training hyperparameters for DNA-Transformer++ models with 2M and 4M parameters. Comma-separated values represent hyperparameter sweeps, with the chosen values indicated in bold.

| Hyperparameters | DNA-Transformer++-2M | DNA-Transformer++-4M |
|---|---|---|
| Embedding Dimension | 256 | 256 |
| Number of Blocks | 4 | 6 |
| Number of Heads | 8 | 8 |
| Up-Projection Ratio | 1.25 | 2.0 |
| Norm Bias and Linear Bias | false | false |
| Context Length | 1,024 | 32,768 |
| Position Embeddings | RoPE | RoPE |
| Learning Rate | 6e-3, 8e-3, **1e-2** | 6e-3, 8e-3, **1e-2** |

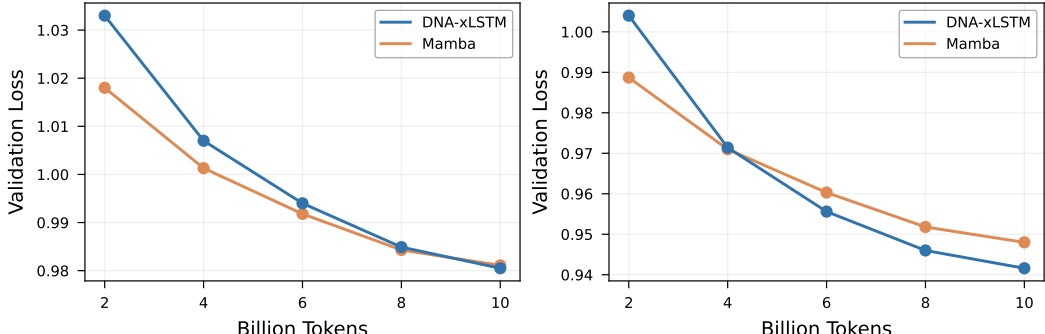

Figure A4: Pre-training of RC-equivariant xLSTM-DNA-PS and Caduceus-PS models with 500k and 2M parameters trained on the human reference genome. Models were trained on 1k context windows using the MLM objective. **Left: MLM losses** (↓) for models with 500k parameters. **Right: MLM losses** (↓) for models in the 2M parameter range. DNA-xLSTM-PS outperforms Caduceus-PS in both settings, with the performance gap widening at larger scales.

are semantically equivalent, as one strand is the RC of the other. Given a strand, $\square$, its RC, $\overline{\square}$, is oriented in the opposite direction with a base conversion from A to T and C to G (Schiff et al., 2024). Shrikumar et al. (2017) show that a data-driven approach to learning the equivalence between reverse-complement sequences can fail, which is why Schiff et al. (2024) propose to enforce RC-equivariance by design, making use of two different inductive biases in the model architecture: PH (Zhou et al., 2022) and PS. For PH models, sequence-to-sequence models — in our case realized by the xLSTM — learn to handle both DNA sequences and their RC during pre-training by applying RC augmentation to the inputs. RC augmentation refers to the process of randomly replacing input sequences by their RCs. For downstream tasks PH models are applied once to the original sequence and once to the RC and eventually outputs are summed:

$$\boldsymbol{Y} = \text{xLSTM}(\boldsymbol{X}) + \text{xLSTM}(\overline{\boldsymbol{X}}). \tag{23}$$

For PS models — we assume models are realized by xLSTM architectures and therefore a block refers to a single mLSTM or sLSTM block — both the DNA sequence and its RC are provided simultaneously to each block in the architecture (for both pre-training and downstream task fine-tuning). Precisely, a joint representation, originating from combining a sequence representation and its RC representation, is split into $\boldsymbol{X} \in \mathbb{R}^{D \times t}$ and $\overline{\boldsymbol{X}} \in \mathbb{R}^{D \times t}$ and fed into the mLSTM or sLSTM block:

$$\left[\boldsymbol{H}, \overline{\boldsymbol{H}}\right] = \left[\text{block}(\boldsymbol{X}), \text{RC}(\text{block}(\text{RC}(\overline{\boldsymbol{X}})))\right]. \tag{24}$$

Notably, for each block the reverse-complement input is built by the RC-function which flips both dimensions of $\overline{\boldsymbol{X}}$ and $[\cdot, \cdot]$ indicates concatenation along the first dimension. Eventually, logits for the input sequence and its reverse complement are combined. For more details, we refer to Schiff et al. (2024).

Table A4: Pre-training hyperparameters of DNA-xLSTM Models from 500k to 4M parameters. Comma-separated values represent hyperparameter sweeps, with the chosen values indicated in bold.

| Hyperparameters | DNA-xLSTM-500k | DNA-xLSTM-2M | DNA-xLSTM-4M |
|---|---|---|---|
| Embedding Dimension | 128 | 256 | 256 |
| Number of Blocks | 5 | 6 | 9 |
| Conv 1D Kernel Size | 4 | 4 | 4 |
| Number of Heads | 4 | 4 | 4 |
| Up-Projection Ratio | 1.25 | 1.0 | 2.0 |
| Bidirectionality | alternating, **blockwise** | alternating, **blockwise** | **alternating**, native, blockwise |
| Norm Bias and Linear Bias | true, **false** | true, **false** | true |
| QKV Projection Blocksize | - | - | 4 |
| m/sLSTM ratio | **[0:1]**, [1:0] | **[0:1]**, [1:0] | [0:1], **[1:0]** |
| Context Length | 1,024 | 1,024 | 32,768 |
| Position Embeddings | None | None | RoPE |
| Optimizer | AdamW $\beta = (0.9, 0.95)$ | AdamW $\beta = (0.9, 0.95)$ | AdamW $\beta = (0.9, 0.95)$ |
| Learning Rate | 6e-3, **8e-3**, 1e-2 | 6e-3, **8e-3**, 1e-2 | 6e-3, **8e-3**, 1e-2 |
| Learning Rate Schedule | Cosine Decay | Cosine Decay | Cosine Decay |
| Learning Rate Warmup Steps | 1,000 | 1,000 | 1,000 |
| Weight Decay | 0.1 | 0.1 | 0.1 |
| Dropout | 0 | 0 | 0 |
| Batch Size | 1,024 | 1,024 | 32 |
| Update Steps | 10,000 | 10,000 | 10,000 |

## C.5 IMPLEMENTATION DETAILS

For both CLM and MLM pre-training we perform 10,000 update steps holding the number of tokens per step constant at $2^{20}$. CLM models are trained using autoregressive next-token prediction. For MLM pre-training, we follow the methodology presented by Devlin et al. (2019), where 15% of the input tokens are masked and the model is tasked to predict the corrupted tokens. Concretely, 80% of the masked tokens are replaced by a special [MASK] token, 10% are replaced by random tokens sampled from the vocabulary and 10% remain unchanged. For MLM settings, we use weight-tied bidirectionality as a default (see Section B.3). For long-context bidirectional modeling, we use unidirectional xLSTM cells and alternate the modeling direction at each block.

To fine-tune pre-trained models on downstream tasks, we follow the framework from Schiff et al. (2024). Pre-trained models are augmented with a task-specific classification head, which is trained on average-pooled activations from a model's final block. During fine-tuning, all model parameters are unfrozen. For the Genomic benchmark, we perform five randomly seeded train-validation splits, fine-tune models for 10 epochs, and use early-stopping on validation performance. Final test results are reported as the mean performance $\pm$ max/min over the 5 seeds on a held-out test set. For the Nucleotide Transformer tasks, we use 20 epochs and 10 seeds. For both the Genomic benchmarks and the Nucleotide Transformer tasks, we performed a hyperparameter search for both PH and PS models over batch sizes $\{64, 128, 256, 512\}$, and learning rates $\{4e-4, 6e-4, 8e-4, 1e-3, 2e-3\}$. The best results for each Nucleotide Transformer task can be found in Table A5.

Table A5: Hyperparameter selection for DNA-xLSTM-PS and DNA-xLSTM-Ph on Nucleotide Transformer tasks. Fine-tuning hyperparameters were chosen based on best scores averaged over ten train-validation splits.

| | DNA-xLSTM-Ph | | DNA-xLSTM-PS | |
|---|---|---|---|---|
| | Learning Rate | Batch Size | Learning Rate | Batch Size |
| *Histone Markers* | | | | |
| H3 | 8e-4 | 128 | 4e-4 | 64 |
| H3K14AC | 6e-4 | 128 | 4e-4 | 64 |
| H3K36ME3 | 6e-4 | 64 | 4e-4 | 64 |
| H3K4ME1 | 8e-4 | 128 | 1e-3 | 128 |
| H3K4ME2 | 6e-4 | 64 | 2e-3 | 512 |
| H3K4ME3 | 8e-4 | 128 | 1e-3 | 512 |
| H3K79ME3 | 1e-3 | 128 | 4e-4 | 64 |
| H3K9AC | 4e-4 | 64 | 1e-3 | 128 |
| H4 | 8e-4 | 64 | 6e-4 | 64 |
| H4AC | 4e-4 | 64 | 1e-3 | 128 |
| *Regulatory Annotation* | | | | |
| Enhancers | 2e-3 | 512 | 2e-3 | 512 |
| Enhancers Types | 2e-3 | 512 | 2e-3 | 512 |
| Promoter All | 4e-4 | 64 | 1e-3 | 128 |
| Promoter No TATA | 1e-3 | 128 | 1e-3 | 128 |
| Promoter TATA | 3e-3 | 128 | 1e-3 | 128 |
| *Splice Site Annotation* | | | | |
| Splice Sites All | 8e-4 | 64 | 2e-3 | 128 |
| Splice Sites Acceptor | 2e-3 | 128 | 2e-3 | 128 |
| Splice Sites Donors | 3e-3 | 128 | 2e-3 | 128 |

# D    PROT-xLSTM: DETAILS AND ADDITIONAL RESULTS

This section provides further details regarding the architecture, training setup, and evaluation metrics for the Prot-xLSTM models. Additionally, we present supplementary results that complement the main findings discussed in Section 4.2. We adopted the experimental protocols outlined in Sgarbossa et al. (2024) to train and evaluate our Prot-xLSTM models. We conducted three key experiments to assess the models' capabilities: **a) protein language modeling** (Section D.1), **b) homology-conditioned protein design** (Section D.2), and **c) protein variant fitness prediction** (Section D.3).

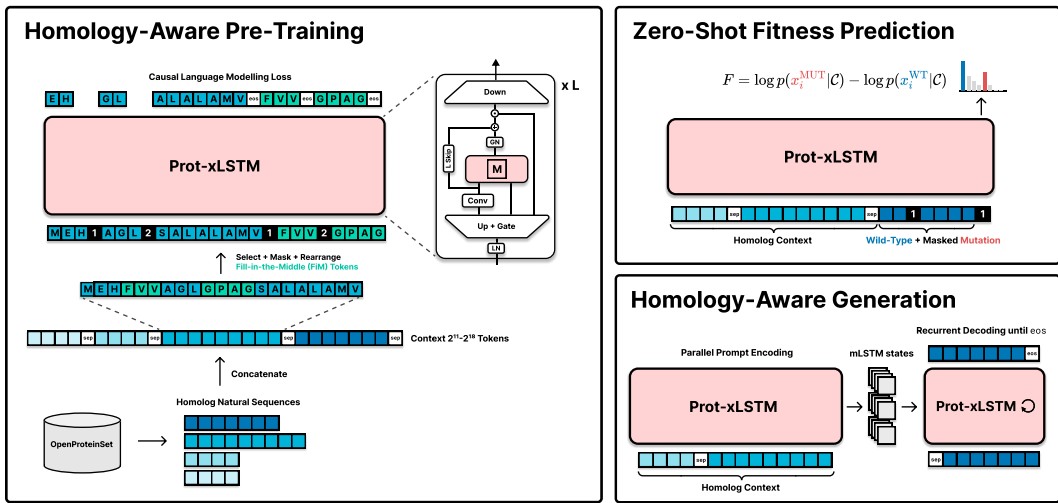

Figure A5: **Prot-xLSTM Framework for Homology-Aware Protein Modeling**. **Left: Homology-Aware Pre-Training** Prot-xLSTM, a 100M parameter causal LSTM model, is pre-trained on homologous proteins using fill-in-the-middle (FIM) augmentations. Homologous protein sequences from the OpenProteinSet are concatenated, creating input contexts ranging from $2^{11}$ to $2^{18}$ tokens. The model is trained with next-token-prediction loss. **Right Top: Zero-Shot Variant Fitness Prediction** Prot-xLSTM predicts protein variant fitness by conditioning on homologous context. Fitness is calculated as the log-likelihood difference between mutant and wildtype sequences and evaluated using the ProteinGym benchmark, where predicted fitness scores are correlated with real-world assay results. **Right Bottom: Homology-Aware Protein Generation** Prot-xLSTM generates proteins by conditioning on sequences from specific families. Generated sequences are scored and compared to real proteins to evaluate their alignment with family-specific properties.

## D.1    HOMOLOGY-AWARE TRAINING

**Data.** The protein language model training data was derived from the filtered OpenProteinSet (Ahdritz et al., 2023), comprising 270k UniClust30 MSA clusters that included a total of 508M sequences and 110B residues. We used the ProtMamba pipeline to construct the training data, which is illustrated in Figure 1 of Sgarbossa et al. (2024) and involved two key steps: (i) creating homology-aware but alignment-free training inputs by concatenating unaligned homologous sequences, and (ii) masking patches of tokens in each sequence and concatenating the unmasked patches at the end of each sequence to train the model autoregressively with the FIM strategy. We also use the train, validation (192 clusters), and test (500 clusters) split provided by ProtMamba.

**Methods and hyperparameters.** We trained two versions of the model: Prot-xLSTM-26M and Prot-xLSTM-102M. The larger model was designed to match the architecture and scale of the original ProtMamba model (ProtMamba-107M) in terms of layer count and embedding size. We optimized the Prot-xLSTM architecture, including block types and positional encodings, on the smaller Prot-xLSTM-26M model, and then applied these optimized architectural choices to the larger Prot-xLSTM-102M model. For comparison, we also trained a smaller ProtMamba model (ProtMamba-28M with an embedding dimension of 512) and implemented a LLaMA-based model (Prot-Transformer++-26M) (Touvron et al., 2023). Both models incorporate Absolute Positional

Encodings (AbsPE) as implemented in ProtMamba for xLSTM blocks, with RoPE applied specifically to the mLSTM blocks. As the sLSTM blocks (or Mamba) lack a $QK$-formulation, RoPE cannot be directly applied to them. The results of the hyperparameter search are provided in The results of the hyperparameter search are reported in Table A7, and the composition of the Prot-xLSTM and Prot-Transformer++ models are reported in Table A6.

Table A6: Hyperparameter space considered for the Prot-xLSTM and Prot-Transformer++ at different sizes. The selected values are marked in bold.

| Hyperparameter | Prot-xLSTM-26M | Prot-xLSTM-102M | Prot-Transformer++-26M |
|---|---|---|---|
| Embedding dimension | 512 | 1024 | 512 |
| Context length | $2^{11},2^{17a}$ | $2^{11-18a}$ | $2^{11}$ |
| Number of blocks | 16 | 16 | 6 |
| m/sLSTM ratio | [0:1], **[1:0]**, [1:7][b] | [1:0] | - |
| Conv 1D kernel size | 4 | 4 | - |
| QKV projection blocksize | 4 | 4 | - |
| Number of heads | 4 | 4 | 8 |
| Up projection dimension | 1024 | 2048 | 2176 |
| Norm bias and linear bias | False | False | False |
| Position embeddings | -, AbPE, **RoPE** | RoPE | RoPE |

[a] Context length was increased during training.
[b] sLSTM blocks at position 1 and 15.

Table A7: Prot-xLSTM hyperparameter search: Training loss comparison across different protein language model architectures after 4B training tokens.

| Model type | #p (M) | Positional Encodings | Train loss |
|---|---|---|---|
| **Mamba** | 27.7 | AbPE | 2.623 |
| **Transformer++** | 26.4 | RoPE | 2.568 |
| **sLSTM** | 25.8 | - | 2.694 |
| | 26.3 | AbPE | 2.688 |
| **mLSTM** | 25.9 | - | 2.569 |
| | 26.4 | AbPE | 2.545 |
| | 25.9 | RoPE | **2.524** |
| | *102* | *RoPE* | ***2.482*** |
| **xLSTM** | 25.9 | - | 2.554 |
| | 26.4 | AbPE | 2.551 |

**Training details.** We trained our models using the ProtMamba pipeline with CLM with the FIM strategy. The pipeline efficiently handles long, concatenated sequences by extending the context length up to $T = 2^{18}$, supported by a context-length scheduling strategy. For the Prot-xLSTM-102M model, we adhered to the ProtMamba protocol, gradually increasing the context length from $2^{11}$ to $T = 2^{18}$, doubling $T$ at each stage when the loss plateaued. In contrast, for the smaller models (Prot-xLSTM-26M and ProtMamba-28M), as recommended in previous work (Devlin et al., 2019; Press et al., 2021), we initially trained with $T = 2^{11}$ for 20B tokens, then switched to $T = 2^{17}$ for an additional 10B tokens. Due to the quadratic scaling of Transformer architectures, Prot-Transformer++-26M was only trained with $T = 2^{11}$, as it could not handle the computational demands of $T = 2^{17}$. Given the substantial computational resources required, we did not fine-tune the training parameters. Instead, we used the default settings established by ProtMamba, which are reported in Table A8.

**Metrics.** During training, we evaluated the next-token prediction capabilities of the models using negative log-likelihood and token perplexity. The perplexity was calculated for four subset of tokens of the concatenated-FIM sequence: the first protein sequence, the last protein sequence the FIM token and the entire concatenated sequence. Once the models were trained we evaluated their performance on the independent test set. We assessed the significance of the test results using pairwise Wilcoxon

Table A8: Hyperparameters for training protein sequence models.

| | |
|---|---|
| Effective batch size[a,b] | 64-1 |
| Optimizer | AdamW $\beta = (0.9, 0.95)$ |
| Learning rate[b,c] | 6e-4 |
| Learning rate scheduler | constant |
| Learning rate warmup steps | 500 |
| Weight decay | 0.1 |
| Dropout | 0 |

[a] Decreased with context size to maintain a fixed total number of tokens per batch. For the larger model, the rule was relaxed for $T >= 2^{16}$ to enable multi-GPU training, with the batch size set to #GPUs.
[b] Prot-Transformer++ was trained on 6 GPUs with an effective batch size of 96 and a learning rate of 9e-4.
[c] Due to unstable training of the larger model at $T = 2^{17}$ and $2^{18}$ the learning rate was reduced to 1e-4.

signed-rank tests on the predictions of the different models. The resulting $p$-values from these tests are presented in Figure A6.

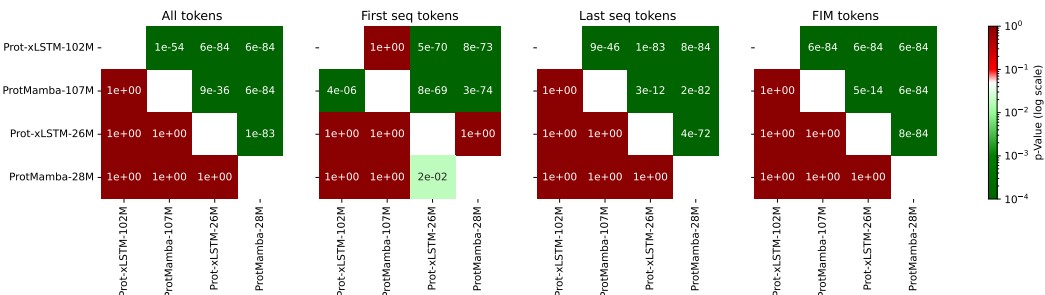

Figure A6: Pairwise one-sided Wilcoxon signed-rank test $p$-values comparing test set perplexities of different protein language models across various token subsets.

## D.2 ICL: HOMOLOGY-CONDITIONED PROTEIN GENERATION

**Experimental setup.** To evaluate the capacity of Prot-xLSTM to autoregressively generate novel protein sequences given a context of known homologs, we follow the protocol outlined in Section 3.4 of Sgarbossa et al. (2024). For a subset of 19 homology clusters from the test set, we generate sequences with contexts consisting of 10, 100, 500, 1000, and $N$ (total number of sequences in the cluster) sequences. For each context length, we generate 100 sequences each with the following parameter combinations of generation temperature ($\tau$), top-$k$, which restricts the output selection to the $k$ most probable tokens, and top-$p$, which limits the output to tokens reaching a cumulative probability $p$: $(\tau, \text{top-}k, \text{top-}p) \in \{(0.8, 10, 0.9), (0.9, 10, 0.95), (1, 10, 0.95), (1, 10, 1), (1, 15, 1)\}$ (Ferruz et al., 2022). This results in a total of 2,500 sequences per cluster.

**Methods compared.** We compare both Prot-xLSTM models to ProtMamba models with a similar number of parameters. Additionally, we generated sequences using PoET (Truong Jr and Bepler, 2023), a Transformer-based, homology-aware protein language model, following the same procedure. All generations were performed with a single A100 64GB GPU. While the xLSTM and Mamba-based models successfully generated sequences across all settings, PoET encountered out-of-memory issues for most clusters when using long contexts: it could handle up to 100 context sequences in 6 clusters, up to 500 in 11 clusters, and only 2 clusters with the full $N$ context sequences. Furthermore, we compared these models with baseline Hidden Markov Models (HMMs) with 100 states, trained on the respective cluster's MSA using the Baum-Welch algorithm (Baum et al., 1970). For each cluster, we generated and evaluated 100 sequences using the HMMs.

**Metrics.** We evaluate the *similarity* of generated sequences by calculating the normalized Hamming distance to the closest natural sequence in the cluster using pairwise Smith-Waterman alignment ($\min d_{\text{nH}}$). Additionally, we measure sequence *similarity to natural homologs* with the HMMER score from a Hidden Markov Model (HMM) trained on the cluster's MSA (Eddy, 2020). To assess the

*foldability*, the generated sequences are also folded using ESMFold (Lin et al., 2023) and assessed by pTM (Evans et al., 2022) and average pLDDT confidence scores (Jumper et al., 2021). To compare these scores with natural sequences, we computed the Kolmogorov-Smirnov test statistic between the scores of 100 natural sequences and the 100 generated sequences with the lowest perplexity. Additionally, we analyzed the diversity of the generated sequences using an adapted version of the #Circles metric to evaluate the number of diverse sequences among the top 100 (lowest perplexity) generated sequences. The #Circles metric, originally proposed by Xie et al. (2023), is a sphere-exclusion-based method designed to quantify the diversity of small molecules in a set using Tanimoto similarity. We adapted this metric for protein sequences by substituting Tanimoto similarity with the Hamming distance and applied a threshold of 0.3 to determine whether two sequences are considered diverse. Finally, we normalized the #Circles metric by the total number sequence and called this the Coverage.

**Results.** We first examined the correlation between protein scores and sequence perplexities. Table A9 shows that the $\min d_{\mathrm{nH}}$, HMMER score, pTM, and pLDDT all correlate with sequence perplexity for both Prot-xLSTM and ProtMamba. For the large models, the average Spearman correlation coefficients are 0.57 and 0.59, respectively. Given this correlation, we limit our subsequent analysis for each model and cluster to the 100 sequences with the lowest perplexity.

Table A9: Score distribution comparison of natural and generated proteins. Average Spearman correlation between model perplexity and sequence scores for sequences generated with Prot-xLSTM and ProtMamba models. Error bars indicate 95% confidence intervals across 19 test clusters.

| | Prot-xLSTM-26M | ProtMamba-28M | Prot-xLSTM-102M | ProtMamba-107M |
|---|---|---|---|---|
| $\min d_{\mathrm{nH}}$ | $0.65^{\pm 0.08}$ | $0.46^{\pm 0.11}$ | $0.68^{\pm 0.10}$ | $0.61^{\pm 0.11}$ |
| HMMER Score | $0.61^{\pm 0.06}$ | $0.55^{\pm 0.08}$ | $0.53^{\pm 0.08}$ | $0.57^{\pm 0.10}$ |
| pLDDT | $0.59^{\pm 0.06}$ | $0.45^{\pm 0.09}$ | $0.52^{\pm 0.07}$ | $0.56^{\pm 0.08}$ |
| pTM | $0.64^{\pm 0.05}$ | $0.54^{\pm 0.07}$ | $0.55^{\pm 0.08}$ | $0.60^{\pm 0.09}$ |

Figure A7 shows the distribution of scores for 100 randomly sampled natural sequences from each cluster as well as the 100 sequences with the lowest perplexity generated by Prot-xLSTM-102M and ProtMamba-107M models for 10 randomly selected clusters. The averages across all 19 evaluated test clusters are shown in Table A10 and in Table Table A11 we report Kolmogorov-Smirnov test statistics between metrics of the 100 natural sequences and the 100 generated sequences. First and foremost, both Prot-xLSTM and ProtMamba outperform the baseline HMM and Transformer-based PoET models in generating higher-quality sequences. Both small and large Prot-xLSTM and ProtMamba models generate a diverse set of sequences that are, on average, longer than their natural homologs. Notably, for the smaller models, Prot-xLSTM generates sequences with properties more closely resembling natural homologs compared to ProtMamba. For the larger models, the difference in quality between Prot-xLSTM and ProtMamba becomes less pronounced. However, Prot-xLSTM sequences remain slightly more similar to natural homologs, as indicated by lower Hamming distances and higher HMMER scores. This increased similarity in the larger Prot-xLSTM model comes at a slight cost to sequence diversity. Finally, all models, except ProtMamba-28M, produce sequences with foldability scores that are highly comparable to those of natural homologs.

Table A10: Score comparison of natural and generated proteins. Average sequence length, scores ($\min d_{\mathrm{nH}}$, HMMER, pLDDT, and pTM), and the coverage of diverse sequences across 19 test clusters for sequences generated with Prot-xLSTM and ProtMamba models. Error bars indicate 95% confidence intervals across clusters

| | Natural Sequences | HMM -12K | PoET -200M | Prot-xLSTM -26M | ProtMamba -28M | Prot-xLSTM -102M | ProtMamba -107M |
|---|---|---|---|---|---|---|---|
| Seq. Len. | $211^{\pm 28}$ | $216^{\pm 25}$ | $330^{\pm 35}$ | $290^{\pm 36}$ | $326^{\pm 43}$ | $286^{\pm 38}$ | $276^{\pm 40}$ |
| $\min d_{\mathrm{nH}} \downarrow$ | $0.51^{\pm 0.04}$ | $0.72^{\pm 0.01}$ | $0.66^{\pm 0.01}$ | $0.55^{\pm 0.05}$ | $0.64^{\pm 0.04}$ | $0.44^{\pm 0.07}$ | $0.56^{\pm 0.03}$ |
| HMMER $\uparrow$ | $96^{\pm 25}$ | $2.29^{\pm 0.88}$ | $89.15^{\pm 17.35}$ | $182^{\pm 56}$ | $122^{\pm 50}$ | $165^{\pm 45}$ | $163^{\pm 45}$ |
| pLDDT $\uparrow$ | $0.81^{\pm 0.03}$ | $0.49^{\pm 0.02}$ | $0.65^{\pm 0.03}$ | $0.79^{\pm 0.04}$ | $0.67^{\pm 0.07}$ | $0.80^{\pm 0.03}$ | $0.80^{\pm 0.03}$ |
| pTM $\uparrow$ | $0.77^{\pm 0.06}$ | $0.23^{\pm 0.01}$ | $0.57^{\pm 0.05}$ | $0.74^{\pm 0.06}$ | $0.54^{\pm 0.10}$ | $0.75^{\pm 0.06}$ | $0.74^{\pm 0.06}$ |
| Coverage $\uparrow$ | $99^{\pm 1}$ | $100^{\pm 0}$ | $91^{\pm 4}$ | $92^{\pm 9}$ | $92^{\pm 10}$ | $87^{\pm 10}$ | $99^{\pm 1}$ |

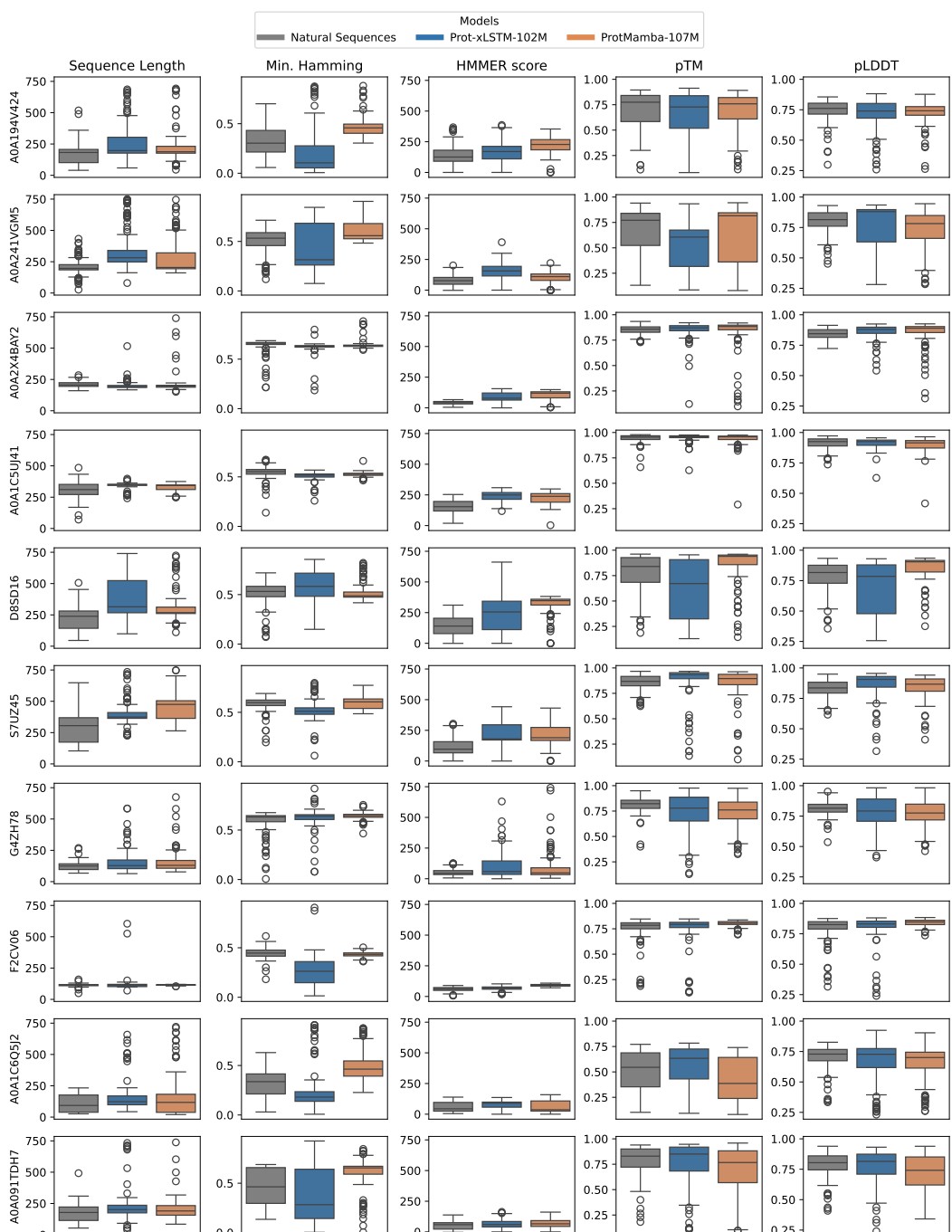

Figure A7: Scores of natural and generated proteins. Boxplots of score distributions for 10 randomly selected clusters evaluated for 100 randomly chosen natural sequences and 100 generated sequences with the lowest perplexity values for large Prot-xLSTM and ProtMamba models.

## D.3 PROTEIN VARIANT FITNESS PREDICTION

**Experimental setup.** We evaluate Prot-xLSTM's ability to predict mutational effects by leveraging its inpainting capabilities from the FIM training objective. This assessment follows the protocol described in Section 3.2 of Sgarbossa et al. (2024) for the ProteinGym DMS substitution benchmark (Notin et al., 2023), which consists of 217 datasets of single and multiple substitutions in protein

Table A11: Average **Kolmogorov-Smirnov statistic** ($\downarrow$) between scores of natural and generated protein sequences with 95% confidence intervals across 19 homology clusters. For three of five metrics, score distributions of Prot-xLSTM-generated sequences are closest to natural sequences.

| | HMM -12K | Prot-xLSTM -26M | ProtMamba -28M | Prot-xLSTM -102M | ProtMamba -107M | PoET -200M |
|---|---|---|---|---|---|---|
| Seq. Len. | $0.37^{\pm0.08}$ | $0.41^{\pm0.09}$ | $0.52^{\pm0.09}$ | $0.40^{\pm0.08}$ | $0.36^{\pm0.08}$ | $0.46^{\pm0.08}$ |
| min $d_{nH}$ | $0.98^{\pm0.01}$ | $0.43^{\pm0.08}$ | $0.60^{\pm0.11}$ | $0.47^{\pm0.09}$ | $0.42^{\pm0.07}$ | $0.80^{\pm0.07}$ |
| HMMER | $0.97^{\pm0.02}$ | $0.57^{\pm0.10}$ | $0.54^{\pm0.11}$ | $0.44^{\pm0.09}$ | $0.49^{\pm0.10}$ | $0.30^{\pm0.07}$ |
| pLDDT | $0.89^{\pm0.05}$ | $0.40^{\pm0.09}$ | $0.68^{\pm0.12}$ | $0.27^{\pm0.05}$ | $0.30^{\pm0.07}$ | $0.67^{\pm0.09}$ |
| pTM | $0.93^{\pm0.04}$ | $0.38^{\pm0.08}$ | $0.72^{\pm0.10}$ | $0.26^{\pm0.05}$ | $0.28^{\pm0.05}$ | $0.65^{\pm0.09}$ |

sequences, allowing comparison with state-of-the-art methods for protein variant fitness prediction. Briefly, for each wild-type sequence, three sets of 200 homologs were obtained by subsampling MSAs following the ColabFold protocol (Mirdita et al., 2022) to be used as context. The context sequences are ordered from the least similar to the most similar one. The wild-type sequence is then concatenated with the context, the mutated residue is masked, and this residue is predicted using the FIM method. Fitness is evaluated as the difference in likelihood between the concatenated sequence with the wild-type and the mutated amino acid and averaged over the triplicate. For multiple mutations, fitness is approximated as the sum of the likelihoods of single mutations.

**Results.** ProteinGym's main metric is the average Spearman correlation between the fitness predictions and the experimental DMS results. Table A12 summarizes reports ProteinGym's main metric, the average Spearman correlation between the fitness predictions and the experimental DMS results, for Prot-xLSTM and several other well-known protein models and the current top of the

Table A12: ProteinGym zero-shot DMS substitution benchmark. The average **Spearman correlation** ($\uparrow$) between predicted fitness scores and experimental measures over 217 DMS assays is shown. While even small Prot-xLSTM models already yield high scores, models that use additional structure tokens, such as SaProt and ProSST, perform best.

| Model Type | Model | Reference | #Params | Spearman $\rho$ |
|---|---|---|---|---|
| Alignment-based | Site-Independant | Hopf et al. (2017) | - | 0.359 |
| | EVE | Frazer et al. (2021) | -[a] | 0.432 |
| | GEMME | Laine et al. (2019) | - | 0.455 |
| Protein language model (PLM) | Tranception L (w/o R) | Notin et al. (2022a) | 700M | 0.374 |
| | VespaG | Marquet et al. (2024) | 3B | 0.458 |
| | ProGen2 XL | Nijkamp et al. (2023) | 6B | 0.391 |
| | ESM-2 | Lin et al. (2023) | 15B | 0.401 |
| Alignment + PLM | MSA-Transformer | Rao et al. (2021) | 100M | 0.432 |
| | Tranception L (w/ R) | Notin et al. (2022a) | 700M | 0.434 |
| | TranceptEVE L | Notin et al. (2022b) | >700M[a] | 0.456 |
| Homology-aware PLM | Prot-xLSTM | Ours | 26M | 0.411[b] |
| | ProtMamba | Sgarbossa et al. (2024) | 28M | 0.360[b] |
| | Prot-xLSTM | Ours | 102M | 0.416[b] |
| | ProtMamba | Sgarbossa et al. (2024) | 107M | 0.415[b] |
| | PoET | Truong Jr and Bepler (2023) | 201M | 0.470 |
| Inverse folding | ESM-IF1 | Hsu et al. (2022) | 142M | 0.422 |
| Structure + PLM | SaProt | Su et al. (2024b) | 35M | 0.407 |
| | ProSST | Li et al. (2024) | 110M | 0.507 |
| | SaProt | Su et al. (2024b) | 650M | 0.457 |

[a] EVE parameters depend on the size of a given MSA.
[b] This work. All other values are retrieved from ProteinGym on 03/11/2024.

# E   CHEM-XLSTM: DETAILS AND ADDITIONAL RESULTS

For chemical sequences, we perform two sets of experiments: **a) unconditional molecule generation** where we follow the experimental protocol of Özçelik et al. (2024). Additionally, we propose a new and more challenging task: **b) conditional generation with ICL**, in which we generate new compounds conditional based on provided in-context compounds.

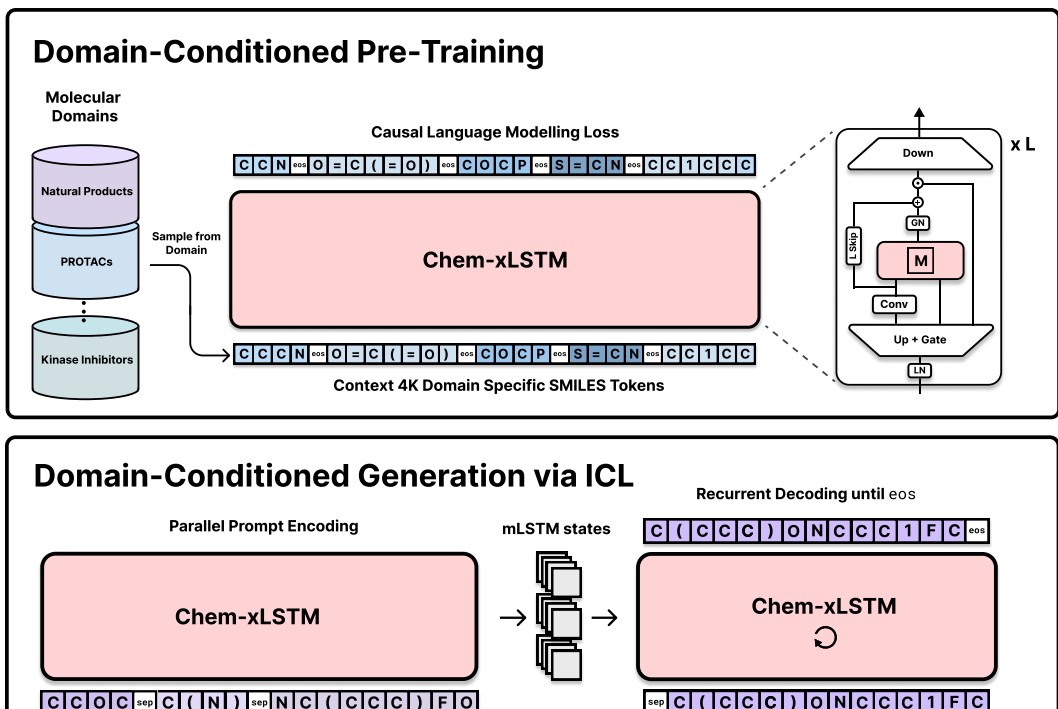

Figure A8: Chem-xLSTM Framework for Domain-Conditioned Molecular Modeling. **Top: Domain-Conditioned Pre-Training.** An autoregressive Chem-xLSTM with 15M parameters is pre-trained on a diverse set of molecular domains. Training contexts of 4,000 tokens are constructed by sampling molecules belonging to specific chemical domains such as PROTACs or natural products. **Bottom: Domain-Conditioned Generation.** At test time, Chem-xLSTM generates novel molecules conditioned on molecular domains. Importantly, the evaluation is performed on unseen domains, requiring the model to rely on its in-context learning capabilities to produce domain-specific molecules. During generation, the domain context can be encoded using the mLSTM's parallel mode and then switch to constant-memory recurrent mode for molecule generation.

## E.1   UNCONDITIONAL MOLECULE GENERATION

*Unconditional molecule generation* is the task of generating valid small molecules without imposing constraints on their characteristics or properties. Generative models aim to learn a general distribution by processing many examples of desirable results. To this end, models are trained on large training sets of arbitrary small molecules without particular conditions or constraints (Segler et al., 2018; Gómez-Bombarelli et al., 2018; Özçelik et al., 2024). Following this approach, we compared the ability of xLSTM and several other models to generate valid and diverse molecules.

**Experimental setup.** For comparability, we aligned our experiments with the setting and dataset of Özçelik et al. (2024). This means that all models are trained to generate molecules as SMILES strings (Weininger, 1988) using a CLM paradigm. The dataset used in (Özçelik et al., 2024) is derived from ChEMBL with a random split in 1.9M training, 100k validation, and 23k test molecules, which have been encoded as SMILES. Before training, all SMILES strings were tokenized using a regular

expression, containing all elements. This results in atoms being represented as one token as well as additional SMILES symbols.

**Methods and hyperparameters.** We compared xLSTM with several other model classes. The first baseline is the default LSTM (Hochreiter and Schmidhuber, 1997) in PyTorch, which includes a forget gate (Gers et al., 1999). This can be considered the direct predecessor of the xLSTM architecture. We also included a variant GPT-2 (Radford et al., 2019) model based on the Transformer architecture (Vaswani et al., 2017) with causal masking. Finally, we included two SSMs in our comparison. On one side, we considered an S4 model with the implementation from Gu et al. (2022), following (Özçelik et al., 2024). On the other side, we incorporated a Mamba model, using the official repository provided with (Gu and Dao, 2024). For our Chem-xLSTM, we used an xLSTM using only mLSTM blocks (Beck et al., 2024). The 15M-parameter model consists of 9 layers with a hidden dimension of 512 and 8 heads. We trained the model for up to 100 epochs with a batch size of 1,024, a context length of 100, a dropout rate of 0.25, and a learning rate of 0.005. All models were trained using the Adam optimizer (Kingma and Ba, 2015) using $\beta = (0.9, 0.999)$, $\epsilon = 1e^{-8}$, and a learning-rate schedule with warm-up and cosine decay. We selected the best model based on the minimum validation loss observed at the end of each epoch. The hyperparameters were manually tuned to match the model parameter count for a fair comparison. Detailed training information and learning curves can be found in Appendix Section E.4.

**Metrics.** We evaluated the models based on next-token perplexity and the Fréchet ChemNet Distance (FCD) (Preuer et al., 2018), a metric adapted from the Fréchet Inception Distance, to assess the similarity between distributions of generated and test-set molecules within the ChemNet logits-space, a model trained on the biological activity of molecules. A lower FCD indicates higher similarity and quality of generated molecules, making it particularly relevant for unconditional molecular design.

Additionally, we report auxiliary metrics in Table A14, capturing aspects such as validity, novelty, diversity, and synthetic accessibility.

- **Validity** reflects the ability to parse SMILES strings without chemical rule violations using RDKit (Landrum, 2013).
- **Uniqueness** ensures distinct molecules by verifying unique InChI keys.
- **Novelty** measures the proportion of generated molecules absent from the training set.
- **Synthetic Accessibility (SA)** quantifies the ease of synthesizing a molecule on a scale from 1 (easiest) to 10 (most difficult) (Ertl and Schuffenhauer, 2009).
- **Diversity** is defined as the fraction of unique Murcko scaffolds among generated molecules.
- **Coverage** is assessed using the #Circles metric (Xie et al., 2023), which counts unique solutions with a Tanimoto similarity threshold of 0.7 (Renz et al., 2024).

These metrics collectively provide a comprehensive evaluation of the syntactic correctness, novelty, and realism of the generated molecules, as well as their alignment with the training distribution. While evaluating unconditional generative models remains challenging (Handa et al., 2023), our focus is on balancing validity, uniqueness, and novelty with computational efficiency.

**Results.** Our proposed Chem-xLSTM model achieved the best results, with the lowest FCD (0.13) and a perplexity (1.68) that is competitive with that of GPT-based models. This indicates that Chem-XLSTM is able to generate realistic chemical structures that match the target distribution well.

All models in our comparison were able to produce valid, unique, and novel molecules. Even though these models have not been optimized for these properties. This is evidenced by the auxiliary metrics surpassing practical thresholds (see Table A14).

## E.2 CONDITIONAL MOLECULE GENERATION WITH IN-CONTEXT LEARNING

Conditional molecule generation with in-context learning (ICL) leverages contextual information to guide the design of novel molecules tailored for specific domains. By incorporating a sequence of molecules as the input, models can conditionally generate new compounds of the same distribution, without the need for fine-tuning.

**Experimental setup.** Similar to the unconditional setup, the input consists of SMILES strings. In the conditional setup, we additionally model sets of molecules from the same molecular domain as a

sequence. Molecules from one molecular domain are serialized and concatenated, separated with the `"."` token. During training, the order of the molecules is permuted to improve generalization and robustness. We construct a novel dataset derived from a variety of molecular domains:

- We consider `natural-products` as domain and utilize the Coconut (Chandrasekhar et al., 2024) as source dataset.
- `Kinase inhibitors`, `withdrawn`, `malaria`, `tool compounds`, `pathogen`, `NIH mechanistic`, `lopac`, `natural product-based probes and drugs`, `zinc tool`, `axon medchem`, `adooq bioactive`, `novartis chemogenetic`, `drug matrix`, `PROTACs`, `covalentIn db`, `DrugBank compounds`, `reframe`, `cayman bioactive` all from the Probes & Drugs portal (Skuta et al., 2017),
- `product molecules` from the reaction dataset USPTO-50k (Lowe, 2012) split into 10 reaction classes.
- The domains `bio`, `diversity`, `green`, `yellow`, `orange`, and `red`, from ZINClick (Levré et al., 2018).
- Active molecules from the domains `BACE`, `BBBP`, `Clintox`, `HIV`, `SIDER`, `Tox21`, `Tox21-10k`, and `Toxcast` from MoleculeNet (Wu et al., 2018).
- Active molecules from 95 bioassays from FS-MOL (Stanley et al., 2021) considered each as separate domain.
- Active molecules from 109 bioassays from PubChem (Kim et al., 2023) considered each as separate domain.
- A subset of active molecules from the BELKA challenge (Quigley et al., 2024) is modeled as a domain.

For the domains that are defined by the active molecules from a particular bioassay, we selected assays with at least 300 active molecules and only use the active compounds. For the dataset each of the total 249 domains is limited to 100,000 compounds, where compounds are selected at random. The final dataset is split at 8:1:1 into train-, validation- and test-domains, sorted by their character length in descending order.

**Methods and hyperparameters.** We benchmark and orient our choices for the model classes as well as hyperparameters based on the unconditional molecule generation results, We consider a context length of 4,096 and adjust batch sizes as well as accumulation steps to accommodate GPU memory constraints. For the S4 model, we were only able to fit a context length of 2,048.

**Metrics.** To evaluate conditional molecule generation we evaluate NTP loss. This metric quantifies how well the model predicts the next token in a sequence, thus assessing whether a model is able to generate molecules from an unseen, and potentially very special, molecular domain given only a few molecules from that domain.

### E.3 ARCHITECTURE AND HYPERPARAMETER SELECTION

Considered and selected hyperparameters for Chem-xLSTM are given in A13.

### E.4 IMPLEMENTATION DETAILS

Unlike Özçelik et al. (2024), we do not backpropagate the loss for `[PAD]` tokens, nor do we interpret them for decoding. We observed that not ignoring `[EOS]` and `[PAD]` token leads to more diversity but is not the standard way of decoding in e.g. NLP. Padding tokens are not typically generated during decoding. They are primarily a pre-processing step to handle batches of data efficiently. In our implementation, we end decoding the SMILES string with the `[EOS]` token. Further, we do not use SMILES augmentation, which could further improve the performance of all architectures.

### E.5 ADDITIONAL RESULTS

Practical thresholds are defined based on several key metrics. First, a high percentage of generated SMILES strings must correspond to chemically valid molecules, with a threshold typically set above

Table A13: Hyperparameter space considered for the Chem-xLSTM at different sizes. The selected values are marked in bold.

| Hyperparameter | Chem-xLSTM-15Mn | Chem-xLSTM-15M-icl |
|---|---|---|
| Number of layers | **9** | **9** |
| Number of heads | **8** | **8** |
| Embedding dimension | **512** | **512** |
| Hidden dimension | **512** | **512** |
| Batch size | 16, **32**, 64, 128 | 16, **32** |
| Proj. factor | **1.3** | **1.3** |
| Learning rate | 1e-4, **2e-4**, 3e-4, 5e-4 | 16, 1e-4, **2e-4**, 3e-4, 5e-4 |
| Optimizer | **Adam**, AdamW | **Adam** |

Table A14: Diversity and correctness metrics for the 15M parameter models for small molecules (SMILES). The table reports the percentage of valid, unique, and novel molecules, the synthetic accessibility (SA), diversity (measured as the percentage of unique Murcko scaffolds), and coverage (#Circles (Xie et al., 2023)) are reported as percentages relative to the total number of generated molecules.

| Model | valid % | unique % | novel % | SA ↓ | diverse % | coverage % |
|---|---|---|---|---|---|---|
| SMILES-LSTM [a] | $90.11^{\pm10.7}$ | $56.72^{\pm3.4}$ | $56.66^{\pm3.6}$ | $2.85^{\pm0.0}$ | $44.71^{\pm1.1}$ | $18.24^{\pm7.2}$ |
| SMILES-GPT [b] | $99.05^{\pm0.5}$ | $62.09^{\pm12.1}$ | $61.81^{\pm12.0}$ | $2.90^{\pm0.0}$ | $48.82^{\pm9.7}$ | $13.44^{\pm0.3}$ |
| SMILES-S4 [c] | $97.48^{\pm0.0}$ | $61.47^{\pm0.0}$ | $61.34^{\pm0.0}$ | $2.86^{\pm0.0}$ | $48.49^{\pm0.0}$ | $13.01^{\pm1.0}$ |
| Chem-Mamba[d] | $91.41^{\pm8.9}$ | $57.75^{\pm3.2}$ | $57.63^{\pm3.8}$ | $2.84^{\pm0.0}$ | $45.65^{\pm7.2}$ | $15.43^{\pm4.6}$ |
| Chem-xLSTM (ours) | $97.08^{\pm0.7}$ | $61.09^{\pm8.9}$ | $60.84^{\pm9.6}$ | $2.83^{\pm0.0}$ | $45.97^{\pm5.5}$ | $14.53^{\pm2.0}$ |

[a] Segler et al. (2018)  [b] Adilov (2021)  [c] Özçelik et al. (2024)  [d] adapted to SMILES in this work

90% to ensure reliability. Additionally, a practical threshold for uniqueness might require that over 80% of the generated molecules are unique, ensuring diversity in the explored chemical space. For novelty, at least 50-70% of the generated molecules should be novel compared to known chemical databases, indicating the model's ability to explore new regions of chemical space. Finally, all models exhibit favorable synthetic accessibility (SA) scores, typically ranging between 2.5 and 5, ensuring that the generated molecules are feasible for synthesis. Further metrics and details are provided in the appendix.

## F COMPUTE DEMAND AND RESOURCES

The experiments were conducted on multiple GPU servers with A100 GPUs. Model training was performed in both single-node and multi-node setups, utilizing 1–8 A100 GPUs per node. Prot-xLSTM-102M training with a context length of $2^{18}$ was completed on a node with 8 H200 GPUs. The largest models were trained across up to four nodes using distributed data parallelism. Some experiments leveraged compute resources provided by EuroHPC Joint Undertaking clusters, including Karolina at IT4Innovations, Leonardo at CINECA, and MeluXina at LuxProvide. The total amount of GPU hours required for the experiments in this paper is approximately 50k.

## G EXTENDED DISCUSSION

In this work, we demonstrated the potential of Bio-xLSTM variants as prime candidates to model biological and chemical sequences. We have provided clarity in two key areas: a) how to tailor xLSTM for biological and chemical sequences, and b) comparing xLSTM-based models to other domain-specific LLMs, showcasing their robust performance in DNA, protein, and chemical sequence modeling tasks. Despite certain limitations, DNA-xLSTM showed strong performance in DNA sequence modeling, excelling in both masked and causal language tasks across different context sizes. In protein modeling, Prot-xLSTM proved particularly effective at handling long-range dependencies, positioning it as a promising tool for generating homologous proteins. In small molecule modeling, Chem-xLSTM achieved the best FCD scores for unconditional generation and demonstrated strong ICL capabilities. Our findings underscore the potential of xLSTM as a prime candidate for foundational models in molecular biology. The models we have introduced, trained, and made available can be used for example to generate rich learned representations for DNA sequences and homology- and chemical domain-conditioned generation of proteins and molecules without the need for fine-tuning.

While Bio-xLSTM has shown strong performance across DNA, protein, and chemical sequence modeling, it has several limitations. The manual hyperparameter selection process, which was due to limited computational resources, may prevent optimal model configurations. We will explore the hyperparameter space further in the future, which might yield even better models. For DNA, the reliance on character-level tokenization might also restrict the performance and scaling to larger context sizes. Also for proteins, amino acid level tokenization without explicit structural information might limit it's performance. The DNA-xLSTM, Prot-xLSTM, and Chem-xLSTM models are currently constrained by the training dataset and their generalizability across organisms and chemical domains needs further exploration. Across all three domains, the training datasets contain biases – whether it is population biases in the genomic data, sequence distribution biases in protein datasets, or chemical exploration biases in molecular datasets. These biases could influence the model's predictions and limit its generalizability in real-world applications. In line with many works, we consider the perplexity metric, for example, next token perplexity, or the related cross-entropy losses as a proxy for performance on downstream tasks. However, this metric might not capture the capacities of biological and chemical language models appropriately. Future work could address these limitations by expanding the training datasets and downstream evaluations of Bio-xLSTM. Finally, assessing Bio-xLSTM's performance in parameter regimes beyond the billion scale remains an open question.

