# OpenReview forum: "Bio-xLSTM: Generative modeling, representation and in-context learning of biological and chemical sequences"
_ICLR.cc/2025/Conference — ICLR 2025 Poster_

### Official Review · Reviewer_QpGg · 2024-10-28

**Soundness:** 2
**Presentation:** 2
**Contribution:** 1
**Rating:** 3
**Confidence:** 5

**Summary:**

This paper presents the Bio-xLSTM framework, which encompasses three specialized models for distinct biological sequences: DNA-xLSTM, Prot-xLSTM, and Chem-xLSTM. DNA-xLSTM introduces reverse-complement equivariant blocks, essential for capturing the symmetry in DNA sequences. Prot-xLSTM is a homology-aware protein language model that employs in-context learning, addressing the variability in protein sequence lengths and context sizes. Chem-xLSTM is designed for SMILES representations of small molecules. Experiments demonstrate that the Bio-xLSTM framework is proficient in generative modeling and representation learning for DNA, proteins, and small molecules.

**Strengths:**

The paper introduces the novel xLSTM architecture, demonstrating its versatility in biological sequence modeling.It incorporates homology awareness in Prot-xLSTM and reverse-complement equivariance in DNA-xLSTM, addressing key challenges in protein and DNA sequence analysis.

**Weaknesses:**

1. **Lack of Motivation**: The paper fails to provide a compelling justification for the use of xLSTM over existing SSM models like Mamba. The authors do not clearly articulate why xLSTM is a better choice for the tasks at hand, which leaves readers without a clear understanding of the advantages it offers over established models.

2. **Lack of Novelty**: The concept of reverse-complement equivariance and the application of post-hoc conjoining (PH) and parameter sharing (PS) within the model architecture have been previously discussed in the literature, as seen in the Caduceus model (Schiff et al., 2024). This reduces the perceived novelty of the current work and may be perceived as a weak contribution to the field.

3. **Results Lack Convincing Evidence**: The manuscript's results section lacks the depth required to fully convince the reader of the proposed xLSTM models' effectiveness. An ablation study would significantly strengthen the paper by demonstrating the impact of the xLSTM components on performance, which is currently not provided.

4. **Redundant Writing**: The paper repetitively introduces the xLSTM architecture and training stategies, which has already been described in the original paper. This redundancy is unnecessary and detracts from the focus on the paper's new contributions and findings.

**Questions:**

1. Could you elaborate on the specific motivations behind choosing xLSTM over other SSM models? What unique advantages or improvements does xLSTM offer in the context of biological sequence design presented in this paper?
2. The performance improvements demonstrated in the paper may not be solely attributable to the xLSTM architecture itself. To convincingly argue that the enhancements are a result of the architecture and not just parameter tuning, the authors could provide a more detailed ablation study.

---

> ### Author Response · Authors · 2024-11-22
> **Response to Reviewer QpGg**
>
> We thank the reviewer for pointing out the proficiency of our method and its versatility in modeling biological sequences!
>
> **W1 and Q1 Motivation**: xLSTM has been shown to be a promising architecture in NLP, offering superior efficiency for long-context inference compared to Transformers. Following the success of adapting NLP architectures for biological and chemical sequences (e.g., HyenaDNA, Caduceus, ProtMamba), we extend xLSTM to these domains. Furthermore, a clear motivation for using xLSTM are a) its higher expressiveness [1] than Mamba,  and b) its efficiency in handling long contexts at inference time [2].
>
> **W2 Novelty**: Our main novelty lies in the adaptation of xLSTM to biological and chemical domains, the thorough assessment of its performance, and providing the pre-trained models to the community. Similar to HyenaDNA, Caduceus or ProtMamba we draw from well-established principles, such as bi-directionality, reverse-complement equivariance, fill-in-the middle, and in-context-learning. Bi-directionality for RNNs via parameter-sharing is well established and has been employed by early large language models such as ELMo [4]. Similarly, both PH invariance and PS equivariance methods for genomics have been developed earlier [5, 6] and Caduceus applied these established principles to causal and masked DNA modeling. We adopt the same methods to compare xLSTM on an equal basis.
>
> **W3 and Q2 Evidence and ablation study**: In this work, we focused on applying xLSTM to biological and chemical sequences, and the original xLSTM paper conducted ablation studies [7] to support its architectural design. Furthermore, across the three domains, we systematically demonstrate that xLSTM matches or outperforms Mamba- and Transformer-based models, showcasing its effectiveness through extensive experiments.
>
>
> **W4: xLSTM formulas as Background**. We include the xLSTM forward pass as Background, which is common practice in papers that apply LSTM or Transformers [e.g. 2,3,4,8]. This part also serves as a basis for the reader and for self-containment.
>
> ---
>
> [1] Merril, W., "The Illusion of State in State-Space Models", International Conference on Machine Learning 41 (2024)
>
> [2] Ross, Jerret, et al. "Large-scale chemical language representations capture molecular structure and properties." Nature Machine Intelligence 4.12 (2022): 1256-1264.
>
> [3] Nguyen, Eric, et al. "Hyenadna: Long-range genomic sequence modeling at single nucleotide resolution." Advances in neural information processing systems 36 (2024).
>
> [4] Peters, M., et. al. "Deep contextualized word representations", NAACL 2018
>
> [5] Mallet V., et. al. "Reverse-Complement Equivariant Networks for DNA Sequences", Advances in Neural Information Processing Systems 34 (2021)
>
> [6] Shrikumar, A., "Reverse-complement parameter sharing improves deep learning models for genomics". BioRxiv, pp. 103663, 2017.
>
> [7] Beck et al., xLSTM: Extended Long Short-Term Memory, NeurIPS, 2024
>
> [8] Kraus et al., xLSTM-Mixer: Multivariate Time Series Forecasting by Mixing via Scalar Memories, arXiv, 2024

---

> > ### Comment · Reviewer_QpGg · 2024-12-01
> >
> > Thank you for addressing my concerns. While the clarifications on the motivations for xLSTM and its advantages are helpful, the overall contribution remains limited due to the lack of novel techniques. I recommend revising the manuscript to strengthen these aspects.

---

### Official Review · Reviewer_zjFj · 2024-11-01

**Soundness:** 3
**Presentation:** 3
**Contribution:** 3
**Rating:** 6
**Confidence:** 3

**Summary:**

The contribution describes application of an LSTM architecture to the modeling of sequential representation of molecules (biomolecules). The motivation for this choice of the architecture is to overcome quadratic scaling of transformers and offer linear scaling and constant memory requirements decoding. Three models are introduced as components of bio-xLSTM suite, specialized in modeling sequential representations of small molecules, proteins, and DNA. The reported results demonstrate performance improvement in certain tasks compared to the alternatives.

**Strengths:**

The paper explores alternatives to the modeling sequential molecular representations in chemistry that are expected to improve scaling and memory requirements. This is a great motivation, considering a) footprint of generative computing and b) its accessibility.

The authors discuss how to tailor the described architecture in relevant tasks.

**Weaknesses:**

The general weakness is that the paper contributes to an oversaturated field but does not offer any breakthrough. The case for LSTM is made by the appeal to their compute requirements, which are not discussed in terms of factual requirements of this study concerning memory bottlenecks, prefactors, scaling, etc. If all the reported results are produced on the same computing resource and performance improvement with LSTM  is marginal, why bother?

**Questions:**

The main motivation for the choice of LSTM is their favorable scaling. There is no scaling analysis in the paper, that clarifies the factual scaling and the observed prefactors. Even with fundamentally improved scaling, prefactors can be prohibitively unfavorable.

The authors provide performance measures in the form of averages and error bars over an ensemble of runs, which is great. At this point, however, it is more meaningful to test if the distributions of performance results are distinguishable instead of comparing averages. The authors do not have to do this exhaustively, but at least for the cases when their models are claimed to be outperforming the alternatives.

---

> ### Author Response · Authors · 2024-11-22
> **Response to Reviewer zjFj**
>
> We thank the reviewer for the constructive feedback on our work! We, too, believe that one of the powerful assets of xLSTM is its linear complexity.
>
> **W1.** We understand the reviewer’s perspective regarding the breakthrough nature of this work. However, we believe our contribution is significant, as it not only advances the state of the art in modeling DNA, proteins, and small molecules but also establishes xLSTM as a consistently strong (and often best) choice across diverse biological modeling scenarios.
>
> We used the following computational resources to train our xLSTM-based models:
> - DNA-xLSTM-2M: was trained in under a day using eight A100 GPUs.
> - Prot-xLSTM-28M: was trained on a single A100 GPU for five weeks.
> - Prot-xLSTM-100M: was trained on four A100 GPUs for six weeks (and additionally for two weeks on eight H200 for $T=2^{18}$).
> - Chem-xLSTM-15M and Chem-xLSTM-15M-icl: were trained in under a day using eight A100 GPUs.
>
> We acknowledge the substantial training resources required. However, re-training these models is unnecessary, as they will be made publicly available. Their efficiency at inference time and ability to handle infinite contexts provide valuable tools for advancing biological modeling across diverse scenarios. We believe the community will benefit from having access to models that provide even marginally improved performance.
>
> Finally, we want to note that comparing xLSTM training times to Transformer- and Mamba-based models is not entirely fair currently, as xLSTM lacks hardware-optimized kernels. Once implemented, we anticipate a 10x speed-up, further enhancing xLSTM's efficiency and practicality.
>
> **Q1: Scaling analysis.** A scaling analysis has already been done in the original xLSTM paper (see Figure 8 of Beck et al., 2024 [1]). For a comprehensive comparison of memory usage and throughput between xLSTM and Transformers with parameter counts similar to those in our work, we refer the reviewer to recent work by Schmied et al. [2].
>
> **Q2**. In addition to the confidence intervals, we now added statistical tests for the performance metrics reported in Tables 1, 2 and 3.
>
> We thank the reviewer for their encouraging words and kindly ask them to check our response and the new version of the manuscript.
>
> ---
>
> [1] Beck, M., et al., xLSTM: Extended Long Short-Term Memory, NeurIPS, 2024
>
> [2] Schmied, T., et. al., A Large Recurrent Action Model: xLSTM enables Fast Inference for Robotics Tasks, ArXiv Preprint (2024)

---

### Official Review · Reviewer_AJhs · 2024-11-02

**Soundness:** 3
**Presentation:** 3
**Contribution:** 2
**Rating:** 6
**Confidence:** 2

**Summary:**

The authors adapt the xLSTM model architecture to the DNA, protein, and chemical informatics space. They compare with SoA models and benchmarks.

**Strengths:**

Generally I think the paper is well written, and the relevant baselines and comparisons across all domains are there.

I think the DNA-xLSTM tasks and benchmarks presented in Table 1 are strong.

Generally, I think the models (parameter sizes, configurations, training data) are comparable across baselines.

**Weaknesses:**

Generally, I feel like this paper is a bit of a grab-bag of computational biology. While the xLSTM architecture is consistent, the additional model additions are quite varied and bespoke.

I’m not sure the large blocks of sLSTM and mLSTM math contribute much to this paper. Do you ever refer back to these equations later in the work?

Typo in header 3 - “BIO-XLSTM: LONGE-RANGE MODELING OF BIOLOGICAL AND CHEMICAL SEQUENCES”

“Hamming distance, HMMER score, and structural scores correlate well with sequence perplexity, with an average absolute Pearson correlation of 0.57 across clusters for the large Prot-xLSTM model” I would not say those correlate well. The R**2 is approx 0.325, which is not a lot of variance explained. Moreover, are these distributions normal? Should you be using spearman?

**Questions:**

“While Transformers have yielded impressive results, their quadratic runtime dependency” - In theory this is true, but it practice, there are more efficient implementations, such as Longformer (Beltagy 202), Linformer (Wang 2020), etc.

For the DNA-xLSTM tasks, to what degree do the PH or PS architecture additions cause a benefit vs just a xLSTM model? Same with the Mamba models (Table 1).

What are the spikes in the validation loss in Figure 3?

For Table 3, I’m always curious about an HMM baseline. How would a HMM of that cluster perform? It’s most certainly going to have many many fewer parameters.

---

> ### Author Response · Authors · 2024-11-22
> **Response to Reviewer AJhs**
>
> We thank the reviewer for their appreciating words and mentioning the comparison across the three domains and the use of strong baselines.
>
> Detailed comments
>
> **W1: Varied additional models**. For pre-training, our comparisons always include at least one Transformer-based architecture, a Mamba-based architecture, and our architecture. Therefore, there is a consistent set of methods compared in all three domains. On downstream tasks, we compare xLSTM to the best-performing methods in the respective domain and task.
>
>
> **W2: xLSTM formulas**.  We prefer to put the xLSTM formulas in the “Background” section for notational clarity and to make the paper self-contained.
>
> **W3: typo**. thanks, corrected.
>
> **W4: Perplexity and protein scores**. The reviewer correctly pointed out that Spearman correlation is more appropriate for assessing the relationship between perplexity and downstream metrics compared to Pearson correlation. Accordingly, we have updated Table A9 to use Spearman correlation in place of Pearson. We have also changed the analysis from “correlate well with” to “correlate with”.
>
> **Q1**: Thanks, we will formulate this more precisely and cite the papers.
>
> **Q2**: We made the same observation as Caduceus [1] during our method development: PH invariance and PS equivariance slightly, but consistently, improve downstream fine-tuning performance. As shown in Table 1 of the Caduceus paper, a non-equivariant variant performs less effectively compared to PH- and PS-equivariant models on the Genomic Benchmarks. Since our goal was to benchmark xLSTM against the strongest Caduceus variants, we limited our comparison to equivariant models.
>
> **Q3**: These are training spikes often observed when training LLMs [2,3], and have been observed for genomic/protein language models [4,5]. They could arise from numerical instabilities in the optimizers.
>
> Nevertheless, since the original submission, we realized that our training setup at long context ($T=2^{17}$) was not optimal concerning stability. Therefore we retrained (i) Prot-xLSTM-26M at $T=2^{17}$ on a single GPU with lr=6e-4 until a total of 30B tokens and (ii) Prot-xLSTM-102M at $T=2^{17}$  on 6 GPU with lr=1e-4 until a total of 60B tokens. We also  further extended the training of Prot-xLSTM-102M to $T=2^{18}$  until a total of 100B tokens. All Prot-xLSTM results have been updated. There are still training spikes in the loss curves, as typical when training LLMs, but they are not as pronounced as before.
>
> **Q4**: We are currently implementing an HMM to provide performance estimates for comparison and aim to complete this analysis before the rebuttal deadline. Additionally, we are working on generating homology-conditioned sequences using PoET [5] to also have a comparison to a Transformer-based model.
>
> We hope that our answers and the updated version could satisfy the reviewer and are kindly asking them to reconsider their assessment and scores.
>
> ---
>
> [1]  Schiff et al., Caduceus: Bi-directional equivariant long-range DNA sequence modeling, ICML, 2024.
>
> [2]  Takase et al, Spike No More: Stabilizing the Pre-training of Large Language Models, arXiv, 2023.
>
> [3]  Nishida et al., Initialization of Large Language Models via Reparameterization to Mitigate Loss Spikes, arXiv, 2024.
>
> [3]  Nguyen et al., Sequence modeling and design from molecular to genome scale with Evo,     bioRxiv, 2024.
>
> [4] Sgarossa et al., ProtMamba: a homology-aware but alignment-free protein state space model, bioRxiv, 2024.
>
> [5] Truong et al., PoET: A generative model of protein families as sequences-of-sequences, NeurIPS, 2023

---

> ### Author Response · Authors · 2024-11-27
> **Further protein model comparisons**
>
> **Addendum W1 and Q4**: we have included additional comparisions for conditional protein generation. We compare Prot-xLSTM against: a) a simple HMM baseline, and b) PoET, a state-of-the-art homology-aware Transformer model [1]. These experimental results are presented in Appendix Section D2 "ICL: Homology-Conditioned Protein Generation" (see Table A11). With these additions, Bio-xLSTM models are consistently benchmarked against both Transformer-based and Mamba-based architectures across all pre-training and downstream experiments.
>
> These results further highlight Prot-xLSTM's strengths compared to simple HMM baselines, as well as, established Transformer models on long-context generation tasks.
>
> ---
>
> [1] Truong et al., PoET: A generative model of protein families as sequences-of-sequences, NeurIPS, 2023

---

### Official Review · Reviewer_R8dF · 2024-11-03

**Soundness:** 4
**Presentation:** 4
**Contribution:** 3
**Rating:** 8
**Confidence:** 3

**Summary:**

The paper introduces Bio-xLSTM, a set of models that is tailored towards modeling of biological sequences. Specifically, the authors apply xLSTMs to different tasks of DNA, protein and small molecular modeling, analyzing the capabilities of xLSTMs in these domains compared to transformer and state-space models. Bio-xLSTM shows remarkable performance across all three domains and achieves state-of-the-art performance at homology aware protein generation.

**Strengths:**

- The usage of xLSTMs in the biological domain is a reasonable approach, particularly in the field of DNA where long context appears to be of importance.
- The paper is well-written, the evaluations and benchmarking is strong and includes reasonable baselines/competitors.
- The overall performance of Bio-xLSTM is strong and Bio-xLSTM is a valuable model for future research in the field.

**Weaknesses:**

- I’m missing an evaluation of the diversity of the generated proteins and small molecules.
- Regarding small molecule generation, I do not really see the benefit of large contexts. Specifically, the unconditional molecule generation uses a context size of 100 tokens. Maybe the authors could explain why the usage of xLSTMs should be beneficial in this domain (I see the point of ICL here, but for unconditional generation there doesn't seem to be any advantage, or?).

**Questions:**

- Do the authors have any explanations why Bio-xLSTM is strong on the histone tasks, but outperformed by NT-v2 500M on the regulatory annotation and splice site annotation tasks (Table A1)? Is there a general difference between the tasks that makes it harder to model it with xLSTMs?

- Did the authors generally observe any patterns where the xLSTM approach is beneficial, and where it might be less effective?

---

> ### Public Comment · ~Yu_Bo1 · 2024-11-12
>
> The reviewer provided valuable insights, but it may be a bit of a misunderstanding to suggest that xLSTM’s advantage over models like Mamba stems primarily from its long-context modeling capabilities. Since the NT benchmark involves sequences of only a few hundred base pairs, it doesn’t fully showcase scenarios where long-context advantages would be most relevant.

---

> ### Author Response · Authors · 2024-11-22
> **Response to Reviewer R8dF**
>
> We thank the author for their positive feedback! Indeed, we, too, think that xLSTM is a strong method for all those biological and chemical domains.
>
> Comments to the specific points raised:
>
> **W1: Diversity**. Indeed our analysis of both of the generated small molecules and proteins was missing an evaluation of the diversity. We have addressed the lack of diversity evaluation by applying the #Circles, a sphere exclusion-based metric, from Xie et al. [1]. It measures the number of unique solutions separated by at least a distance threshold (Tanimoto similarity for small molecules, adapted to Hamming distance for proteins). Results have been added to Tables A10 (proteins) and A3 (small molecules). We found out all protein language models were able to generate a diverse set of sequences: over 85% of sequences have at least 0.3 minimum Hamming distance to their closest neighbor. For the molecule generators, this proportion of diverse sequences is close to 15% for all generators.
>
> **W2: Benefit of large context for chemical data**. The reviewer is right that the benefit of long-context models exists only in conditional generation, where it enables in-context learning (ICL) to condition molecule generation using example molecules. Long contexts are unnecessary for unconditional generation, which is why they were not used. The purpose of the unconditional generation experiments was to demonstrate that xLSTM performs on par with other architectures in this domain and to serve as a baseline before introducing the novel conditional generation approach with ICL.
>
>
> **Q1: Performance difference for task types**. We are currently investigating the differences between task types that might explain the observed performance variation. Histone binding involves recognizing specific patterns in DNA where proteins can bind, which is well-suited for xLSTMs. In contrast, regulatory annotation and splice site detection are more complex tasks that likely require the model to store and process more extensive information, favoring larger models.
>
> This trend is consistent with observations from HyenaDNA and Caduceus, both of which show similar performance patterns. Notably, HyenaDNA’s analysis (Appendix Table A.6) [2] included a small GPT model with a parameter count comparable to HyenaDNA, which also underperformed relative to larger Transformers on regulatory annotation and splice site detection. These results suggest that the performance differences arise from disparities in model capacity rather than differences between linear methods and Transformers.
>
>
> **Q2: General patterns when xLSTM performs well**. sLSTM excels in tasks requiring state tracking [3], making it effective in domains like code and mathematics, and, as we show, for short-context DNA sequence modeling. mLSTM, with its expressive gating mechanisms, provides enhanced memory capacity and performs well in protein modeling. xLSTM, due to its compressive nature, appears to learn robust abstractions, improving generalization under domain shifts. For instance, in molecule generation, xLSTM outperforms Transformers on unseen molecular domains and demonstrates strong robustness to length extrapolation without retraining, as shown in the original xLSTM paper.
> Transformers, however, remain superior for tasks requiring exact copying of information, such as the MQAR experiment in the xLSTM paper and recent works [4,5]. These observations suggest opportunities for future work in hybrid models that combine the strengths of xLSTM, sLSTM, and Transformers.
>
> ---
>
> [1] Xie, Y. et al., How Much Space Has Been Explored? Measuring the Chemical Space Covered
>      by Databases and Machine-Generated Molecules. ICLR, 2023
>
> [2] Nguyen. E. et.al., HyenaDNA: Long-Range Genomic Sequence Modeling at Single Nucleotide Resolution. NeurIPS, 2023
>
> [3] Grazzi, R. et. al., Unlocking State-Tracking in Linear RNNs Through Negative Eigenvalues
>
> [4] Waleffe et al., An Empirical Study of Mamba-based Language Models, arXiv, 2024
>
> [5] Jelassi et al., Repeat after me: Transformers are better than state space models at copying, arXiv, 2024

---

### Meta-Review · Area_Chair_eWRB · 2024-12-22

**Metareview:**

This paper introduces Bio-xLSTM, a set of models leveraging the xLSTM architecture for modeling biological and chemical sequences across three domains: DNA, proteins, and small molecules. The architecture addresses limitations of Transformers in terms of quadratic scaling and memory efficiency, offering linear runtime dependency and constant-memory decoding. The authors adapt the architecture with domain-specific enhancements, such as reverse-complement equivariance for DNA and homology awareness for proteins. Evaluations demonstrate competitive performance in generative modeling and representation learning tasks.

**Additional Comments On Reviewer Discussion:**

The reviewers noted strengths such as the ability to handle long contexts efficiently and the comparisons against state-of-the-art models. Concerns were raised about the lack of novelty, particularly regarding the reuse of established techniques such as equivariance and parameter sharing. The authors responded to these concerns by emphasizing the contribution of adapting xLSTM to new domains and providing additional results, including diversity metrics and comparisons with strong baselines like HMMs and PoET. There was lack of consensus on the paper with some reviewers maintaining reservations about the novelty and the limited evaluation on long-range experiments.

---

### Decision · Program_Chairs · 2025-01-22

Accept (Poster)